# Alkalinity sources in the Dutch Wadden Sea

Mona Norbisrath[1,2,3], Justus E. E. van Beusekom[1], & Helmuth Thomas[1,2]

[1]Institute of Carbon Cycles, Helmholtz-Zentrum Hereon, Geesthacht, 21502, Germany

[2]Institute for Chemistry and Biology of the Marine Environment (ICBM), Carl von Ossietzky University Oldenburg, Oldenburg, 26129, Germany

[3]now at: Department of Marine Chemistry and Geochemistry, Woods Hole Oceanographic Institution, Woods Hole, MA, 02543, USA

*Correspondence to*: Mona Norbisrath (mona.norbisrath@gmail.com)

## Abstract

Total alkalinity (TA) is an important chemical property playing a decisive role in the oceanic buffering capacity of $CO_2$. TA is mainly generated by weathering on land, and by various anaerobic metabolic processes in water and sediments. The Wadden Sea, located in the southern North Sea is hypothesized to be a source of TA for the North Sea, but quantifications are scarce. This study shows observations of TA, dissolved inorganic carbon (DIC), and nutrients in the Dutch Wadden Sea in May 2019. Along several transects, surface samples were taken to investigate spatial distribution patterns and to compare them with data from the late 1980s. A tidal cycle was sampled to further shed light on TA generation and potential TA sources. We identified the Dutch Wadden Sea as a source of TA and estimated an export of 6.6 Mmol TA per tide to the North Sea. TA was generated in the sediments with deep pore water flow during low tide enriching the surface water. A combination of anaerobic processes and $CaCO_3$ dissolution were potential TA sources in the sediments. We deduce that seasonality and the associated nitrate availability in particular influence TA generation by denitrification, which is low in spring and summer.

## 1 Introduction

As the regulator of the ocean carbon dioxide ($CO_2$) sink, total alkalinity (TA) is of increasing scientific interest and is investigated worldwide in the so called "Anthropocene" (Abril and Frankignoulle, 2001;Bozec et al., 2005;Chen and Wang, 1999;Dickson, 1981;Middelburg et al., 2020;Norbisrath et al., 2022;Renforth and Henderson, 2017;Thomas et al., 2004;2009;Sabine et al., 2004). The "Anthropocene" describes the current era of our planet, when environmental changes, driven by humans, have become identifiable in geological records (Zalasiewicz et al., 2010;Crutzen, 2002). One of the most threatening changes for our climate is the anthropogenic driven increase in atmospheric greenhouse gases (GHG), such as $CO_2$. To counteract the increasing atmospheric $CO_2$ concentrations and the ongoing climate warming, a combination of several pathways is needed. Beside a strict reduction of $CO_2$ emissions, also net-negative emissions are required, which capture the atmospheric $CO_2$ and store it either based on land or in the ocean (e.g., Keith et al., 2006;Matthews and Caldeira, 2008;Zhang

et al., 2022). The climate and the increasing atmospheric $CO_2$ content is also naturally regulated by the open ocean, and around
a quarter of the global anthropogenic $CO_2$ emissions are already removed by it (Friedlingstein et al., 2022). The carbon storage
capacity of the North Sea is an important atmospheric $CO_2$ sink as it exports the absorbed $CO_2$ in the deep layers of the Atlantic
Ocean where it is stored on longer time scales (Borges et al., 2005;Bozec et al., 2005;Burt et al., 2016;Brenner et al., 2016;Hu
and Cai, 2011;Schwichtenberg et al., 2020;Thomas et al., 2004;2009). Two important aspects of the oceanic climate regulation
are the oceanic circulation and TA. TA, primarily consisting of bicarbonate and carbonate, is generated by chemical rock
weathering (Suchet and Probst, 1993;Meybeck, 1987;Berner et al., 1983), and in various stoichiometries by calcium carbonate
($CaCO_3$) dissolution and anaerobic metabolic processes, such as denitrification, which is the reduction process of nitrate to
dinitrogen gas in the nitrogen cycle (Hu and Cai, 2011;Wolf-Gladrow et al., 2007;Chen and Wang, 1999;Brewer and Goldman,
1976). Since TA, $CO_2$ uptake and its export to the deep ocean are mainly disentangled in the open ocean, TA and the oceanic
circulation interact closely in highly active and shallow ocean areas such coastal zones and continental and marginal shelves.
In these shallow areas, TA is susceptible to changes due to various metabolic processes and the influence of adjacent zones
like rivers, estuaries, marshes, and tidal flats (e.g., Norbisrath et al., 2022;2023;Wang et al., 2016;Voynova et al., 2019). A
previous study by Norbisrath et al. (2022) showed that an enhanced riverine, metabolic alkalinity would lead to increasing
$CO_2$ absorption in the coastal zones of the North Sea, highlighting the need to further investigate TA regulation in adjacent
zones of coastal oceans.
Coastal zones, which are the direct interface between most, if not all, compartments of the Earth system (i.e., atmospheric,
terrestrial, aquatic, and oceanic) and human societies, appear particularly vulnerable to environmental and climate change
(Glavovic et al., 2015). This holds true for the Wadden Sea, the shallow, coastal sea along an approximately 500 km coastline
of the Netherlands, Germany, and Denmark, in the southern North Sea, which is declared as an UNESCO world natural heritage
site since 2009. Most of the Wadden Sea is located between the protecting barrier Islands and the Mainland, which makes it
the world's largest uninterrupted stretch of tidal flats with multiple tidal inlets (Fig. 1). Due to the topography, the Wadden
Sea is a highly dynamic ecosystem with influences from the mainland and the North Sea (Hoppema, 1993;Postma, 1954;van
Raaphorst and van der Veer, 1990). Driving forces of the biogeochemical dynamics in the Wadden Sea are nutrient imports
by rivers and high suspended particulate matter (SPM) and organic matter (OM) imports from the North Sea (van Beusekom
et al., 2019;van Beusekom et al., 2012;Postma, 1954). Physical sources of variability in the Wadden Sea are oceanic driven
wind, waves, and tidal currents, as well as the counterclockwise circulation of the North Sea (Elias et al., 2012). Large tidal
amplitude and currents in conjunction with shallow water depths allow for vertical water column mixing and an exchange
between the pelagic and benthic realms including deep pore water exchange (Røy et al., 2008). The strong tidal currents also
impact the biogeochemistry of the North Sea (Postma, 1954), as they cause an exchange of water between the North Sea and
the Wadden Sea and play an important role in the import of particulate matter from the North Sea (Burchard et al., 2008).
Previous studies identified the Wadden Sea as a TA source for the North Sea with a loading between 39 Gmol yr$^{-1}$
(Schwichtenberg et al., 2020) and 73 Gmol yr$^{-1}$ (Thomas et al., 2009). Both studies suggested the entire Wadden Sea as one of
the most important TA sources of the carbon storage capacity for the North Sea. Burt et al. (2016) highlighted the importance
of coastal TA production for regulating the buffer system in the North Sea, and suggested denitrification as the major TA
source. Due to the strong connection between the North Sea and the Wadden Sea, a better understanding of TA generation in
the latter is required. Here, we focus on the Dutch Wadden Sea that has been well-studied during the past decades (Hoppema,
1990, 1991, 1993;De Jonge et al., 1993;Elias et al., 2012;Ridderinkhof et al., 1990;Postma, 1954;van Beusekom et al.,
2019;Schwichtenberg et al., 2017). In particular Hoppema (1990);(1993) observed the spatial and temporal variability of TA
in May in the late 1980s, which we compare with our observed transect data to detect potential differences over the last 30
years. In addition, we further discussed potential TA sources in the Dutch Wadden Sea.
**2 Methods**
**2.1 Study site and sampling**
This study is based on samples collected on a research cruise (LP20190515) in the Dutch Wadden Sea (Frisian Islands) on RV
*Ludwig Prandtl* in May 2019 (Fig. 1). We collected water samples in the Wadden Sea starting at Harlingen, through the Vlie
Inlet along the islands Vlieland and Terschelling, through the Ameland Inlet to Ameland Island, from there on via the Frisian
Inlet to Lauwersoog, and around Schiermonnikoog Island via the Ems-Dollard Inlet to Emden. In addition, we sampled a half
tidal cycle during ebb tide (from high tide to low tide) on 21 May 2019 (Table B2). To set the range of ebb tide data in relation,
we also sampled a half tidal cycle during flood tide (from low tide to high tide) on 23 May 2019 for comparison. Both half
tidal cycles were sampled as an anchor station in the waterway at the western side of Ameland in the Ameland Inlet on each
day.
Nearly half-hourly, we collected discrete surface (1.2 m depth) water samples with a bypass from the onboard flow-through
FerryBox system, which also provided essential physical parameters such as salinity with an accuracy of 0.02 (PSU) and
temperature with an accuracy of 0.1 °C (Petersen et al., 2011). The FerryBox was cleaned and the system checked prior to the
cruise, and salinity is occasionally checked using discrete samples, which is considered sufficient for gradients in near-shore
investigations (pers. comm. Y. Voynova). We complemented our salinity and temperature data with data from three
Rijkswaterstaat stations (Dantziggat, Terschelling 10, and Vliestroom; Table B3), which were close to our stations.
For TA and DIC measurements we sampled water with overflow into 300 mL BOD (biological oxygen demand) bottles and
preserved them with 300 µL saturated mercury chloride solution ($HgCl_2$) to stop biological activity. Each BOD bottle was
filled without air bubbles and closed by using a ground-glass stopper coated in Apiezon® type M grease and a plastic cap. The
samples were stored in a cool dark environment until measurements in the lab.
Water for nutrient samples was filtered through pre-combusted (4 h, 450 °C) GF/F filters and the filtrate was stored frozen in
three 15 mL Falcon tubes for triplicate measurements in the lab.
To determine the total carbon (C), organic carbon ($C_{org}$) and nitrogen (N) concentrations in SPM and associated $C_{org}$:N ratios,
we used pre-combusted (4 h, 450 °C) GF/F filters, which were dried after sampling at 50 °C to remove all humidity and were
stored frozen afterwards until measurement.

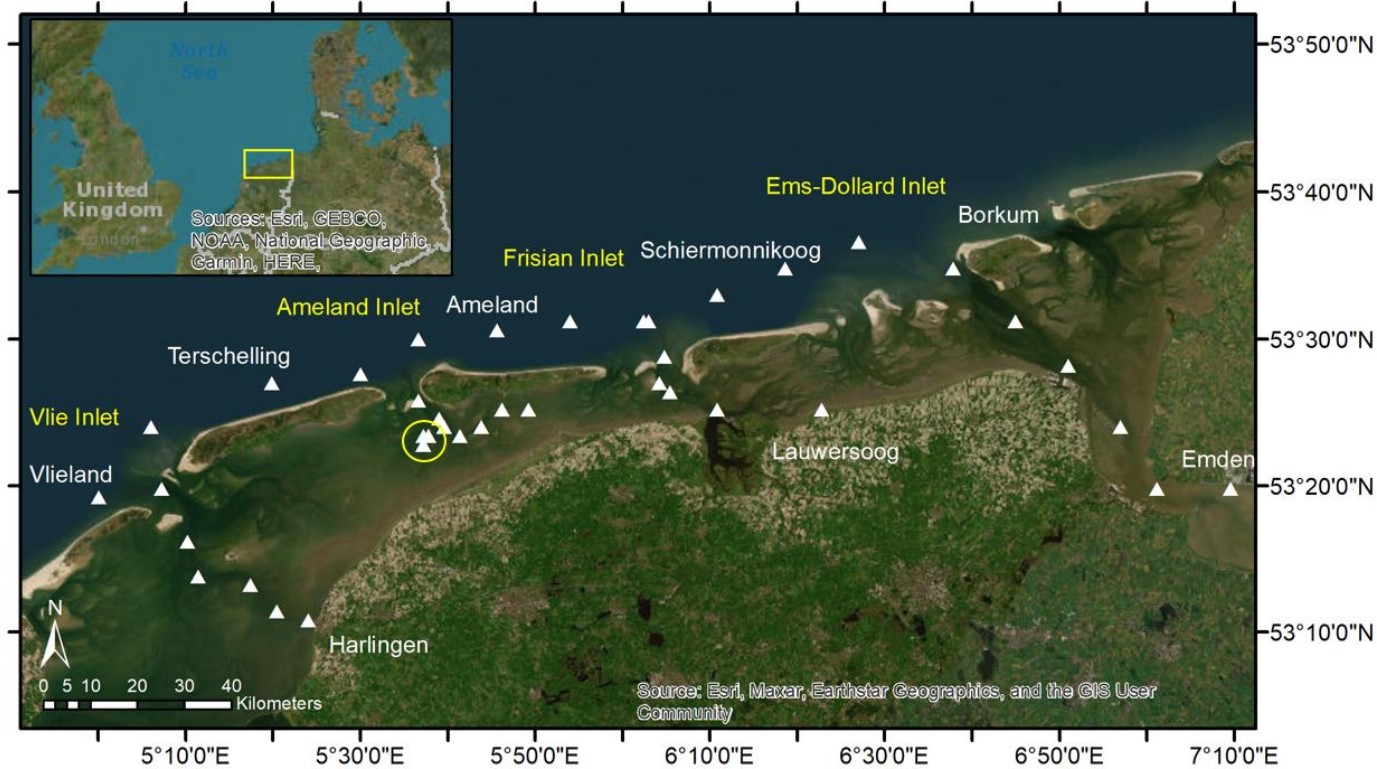


**Figure 1** Sampling site in the Dutch Wadden Sea. The sampling stations around the Frisian Islands in May 2019 are visualized
with white triangles. The yellow circle highlights the anchor stations for the tidal cycle sampling in the Ameland Inlet on two
days. During the sampling day from low tide to high tide, we had two samples that we took slightly more western due to
drifting. The island and city names are shown in white, the inlets in yellow. The tidal flats and sedimentary structures are well
visible between the barrier islands and the mainland.
**2.2 Carbon species analyses**
The parallel analyses of TA and DIC were carried out in March 2020 by using the VINDTA 3C (Versatile INstrument for the
Determination of Total dissolved inorganic carbon and Alkalinity, MARIANDA - marine analytics and data), which measures
TA by potentiometric titration and DIC by coulometric titration both with a measurement precision < 2 µmol kg$^{-1}$ (Shadwick
et al., 2011). Certified reference material (CRM batch # 187) provided by Andrew G. Dickson (Scripps Institution of
Oceanography) was measured before and after the samples and used to ensure a consistent calibration of both measurements.
The calcite and aragonite saturation states ($\Omega$), the pH, and the seawater partial pressure of $CO_2$ ($p$CO$_2$) were computed with
the $CO_2$SYS program (Lewis and Wallace, 1998), using the measured parameters TA and DIC, and salinity, temperature,
silicate and phosphate as input variables, together with the dissociation constants from Mehrbach et al. (1973), as refit by
Dickson and Millero (1987). Reported calculation uncertainties are ± 0.0062 for pH (Millero et al., 1993), ± 4.9 % for the
aragonite saturation state and ± 3.5 % for $p$CO$_2$ (Orr et al., 2018).

**2.3 Nutrient analyses**

The nutrients were measured with a continuous flow automated nutrient analyzer (AA3, SEAL Analytical) and a standard colorimetric technique (Hansen and Koroleff, 2007) for nitrate ($NO_3^-$), nitrite ($NO_2^-$), phosphate ($PO_4^{3-}$), and silicate (Si), and a fluorometric method (Kérouel and Aminot, 1997) for ammonium ($NH_4^+$) (Grasshoff et al., 2009). The nutrient samples were measured against Eurofins reference materials VKI SW4.1B (for NOx, $NO_2$ and $NH_4$) and VKI SW4.2B (for Si and $PO_4$) in July 2019. The maximum standard deviations were 0.322 µmol $L^{-1}$ for $NO_3^-$, 0.014 µmol $L^{-1}$ for $NO_2^-$, 0.081 µmol $L^{-1}$ for $NH_4^+$, 0.014 µmol $L^{-1}$ for $PO_4^{3-}$ and 0.165 µmol $L^{-1}$ for Si.

For the $C_{org}$ determination, filters were acidified with 1N HCl and dried overnight to remove all inorganic carbon content. Filters were measured with a CHN-elemental analyzer (Eurovector EA 3000, HEKAtech GmbH) in the Institute of Geology, University Hamburg, and calibrated against a certified acetanilide standard (IVA Analysentechnik, Germany). The standard deviations were 0.05 % for carbon and 0.005 % for nitrogen.

**2.4 Data analyses**

The data analyses were performed by using RStudio Version 1.3.1073 © 2009-2020 RStudio, PBC. The linear regression Model II was performed by using the "lmodel2" R package, and the plots were created with the "ggplot2" R package.

**3 Results**

**3.1 Spatial parameter distribution**

To investigate the spatial distribution of TA in the Dutch Wadden Sea and compare its general status with earlier studies (in particular Hoppema, 1990), we observed TA and related parameters in surface water along a transect from the coastal mainland towards the North Sea.

The temperatures varied between 12 and 16 °C with higher temperatures towards the coastal mainland (Fig. 2a). We identified two main sub regions based on the salinity values. First the Ems-Dollard Inlet, which showed salinities lower than 28 and with the minimum value of 20.25 at the most upstream station. And second, around Ameland Island and the remaining of our investigated region in the Dutch Wadden Sea with salinities showing only smaller variations varying from 28 to 33 (Fig. 2b). Spatial transect TA concentrations ranged from 2332 µmol TA $kg^{-1}$ to 2517 µmol TA $kg^{-1}$. We observed lower concentrations on the North Sea side of the Frisian Islands with somewhat higher concentrations around Ameland (Fig. 2c). In contrast to the North Sea side, the values were higher ($> 2380$ µmol TA $kg^{-1}$) in the Wadden Sea. In the Ems-Dollard Inlet, the concentrations were even higher, with values up to 2517 µmol TA $kg^{-1}$ at the most upstream station.

Silicate (Si) showed higher concentrations in the Wadden Sea and lower ones towards the North Sea (Fig. A1a). Highest concentrations were observed at the coastal mainland and in the Ems-Dollard Inlet. Silicate concentrations ranged between 0.3 and 56.3 µmol Si $L^{-1}$. Both, the calcite and aragonite saturation states ($\Omega$) were supersaturated in the entire study region.

Saturation state values ranged from 2.3 to 4.6 for calcite (Fig. A1b), and from 1.4 to 2.8 for aragonite (Table B3). Highest
values were observed at the North Sea side of the barrier islands, and lowest values near Harlingen and in the Ems-Dollard
Inlet. Like the calcite and aragonite saturation states, the pH values were higher in the North Sea, and lower in the Wadden
Sea and near the coastal mainland (Fig. A1c). The pH values ranged from 7.86 to 8.19, and lowest values were observed near
Harlingen and in the Ems-Dollard Inlet. The nitrate ($NO_3^-$) concentrations were in a low range ($< 3\ \mu mol\ NO_3^-\ L^{-1}$) throughout
the study region. Higher concentrations ($< 6\ \mu mol\ NO_3^-\ L^{-1}$) were observed only at a few stations close to land, and maximum
concentrations ($< 38\ \mu mol\ NO_3^-\ L^{-1}$) were observed in the Ems-Dollard Inlet (Fig. A1d). DIC concentrations ranged from 2097
$\mu mol\ DIC\ kg^{-1}$ to 2430 $\mu mol\ DIC\ kg^{-1}$ (Fig. A1e). DIC values showed a similar pattern as TA values, with higher concentrations
near the coastal mainland and in the Ems-Dollard Inlet, and decreasing concentrations toward the North Sea, where DIC
reached minimum values.

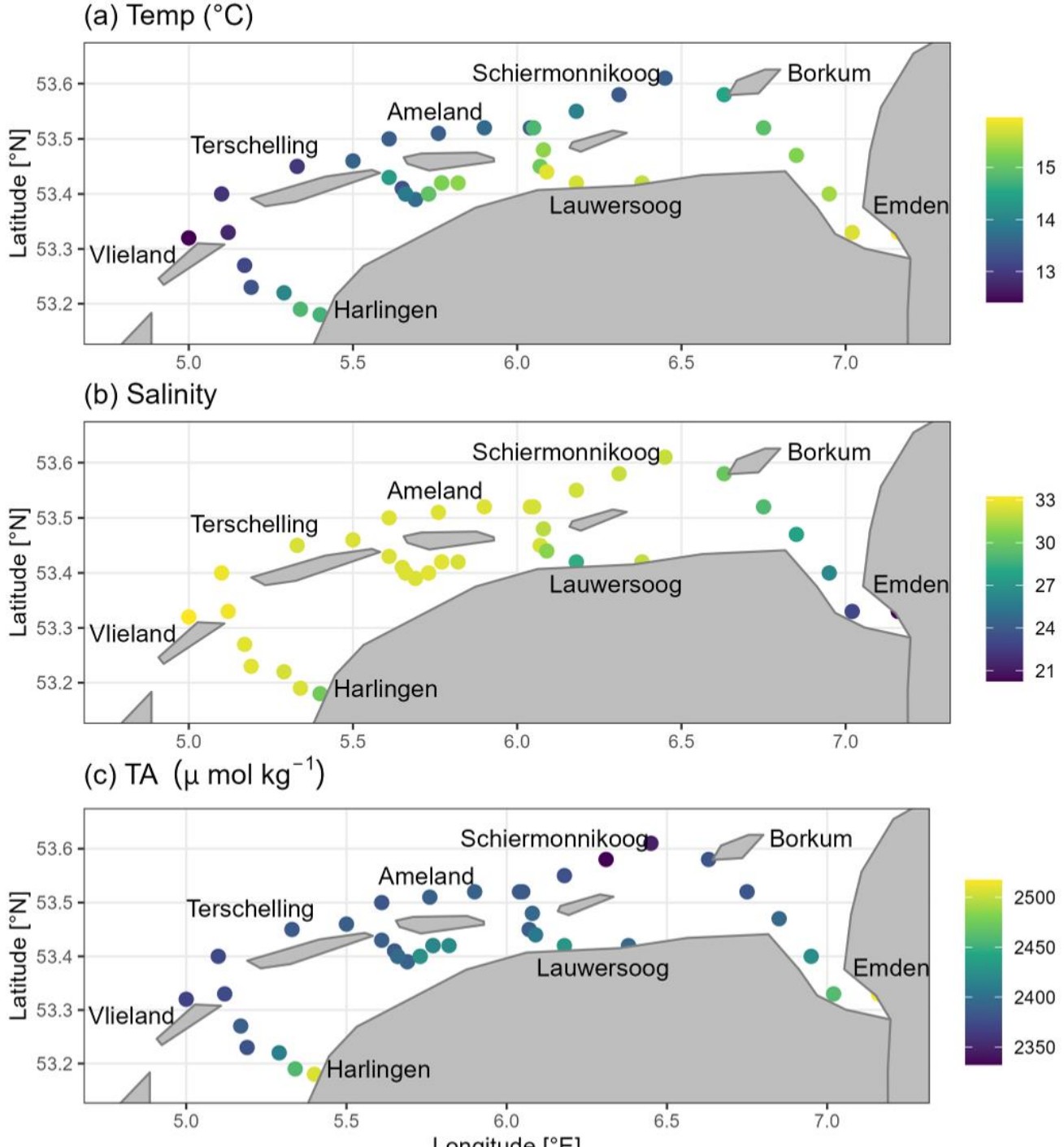


**Figure 2** Spatial distribution of a) temperature (°C), b) salinity (PSU), and c) total alkalinity (TA; µmol kg⁻¹) from surface
water samples in May 2019.

Compared to the other transects of this study region, the strong influence of the inner Ems Estuary is visible at the most
upstream station in the Ems-Dollard Inlet, showing lowest salinity, lowest pH and calcite saturation state values, and highest
values of TA, DIC, nitrate, silicate and phosphate. The outer side of the Vlie Inlet reflects the North Sea conditions with lower
temperatures and higher salinities. The North Sea impact is also visible in the mixing plot between TA and salinity (Fig. 3).
Statistical significant linear mixing behavior was observed in the transect through the Ems-Dollard Inlet ($R^2 = 0.81$) and
through the Vlie Inlet ($R^2 = 0.77$), where TA concentrations decreased with increasing salinities from the mainland towards
the North Sea (Fig. 3). Whereas in the Ems-Dollard Inlet mixing is dominated by riverine water with high TA concentrations,
the mixing in the Vlie Inlet showed a more prominent mixing of Wadden Sea and North Sea water. The TA concentrations in
the Vlie Inlet and around Ameland, both at the North Sea side (Ameland NS) and the Wadden Sea side (Ameland WS) were
higher than the TA concentration computed for the salinity end-member in the Ems-Dollard Inlet, suggesting the Dutch
Wadden Sea as a source of TA (Fig. 3). Both the Ameland NS and WS data clearly indicated a non-conservative behavior with
a range of TA concentrations at near constant salinities.

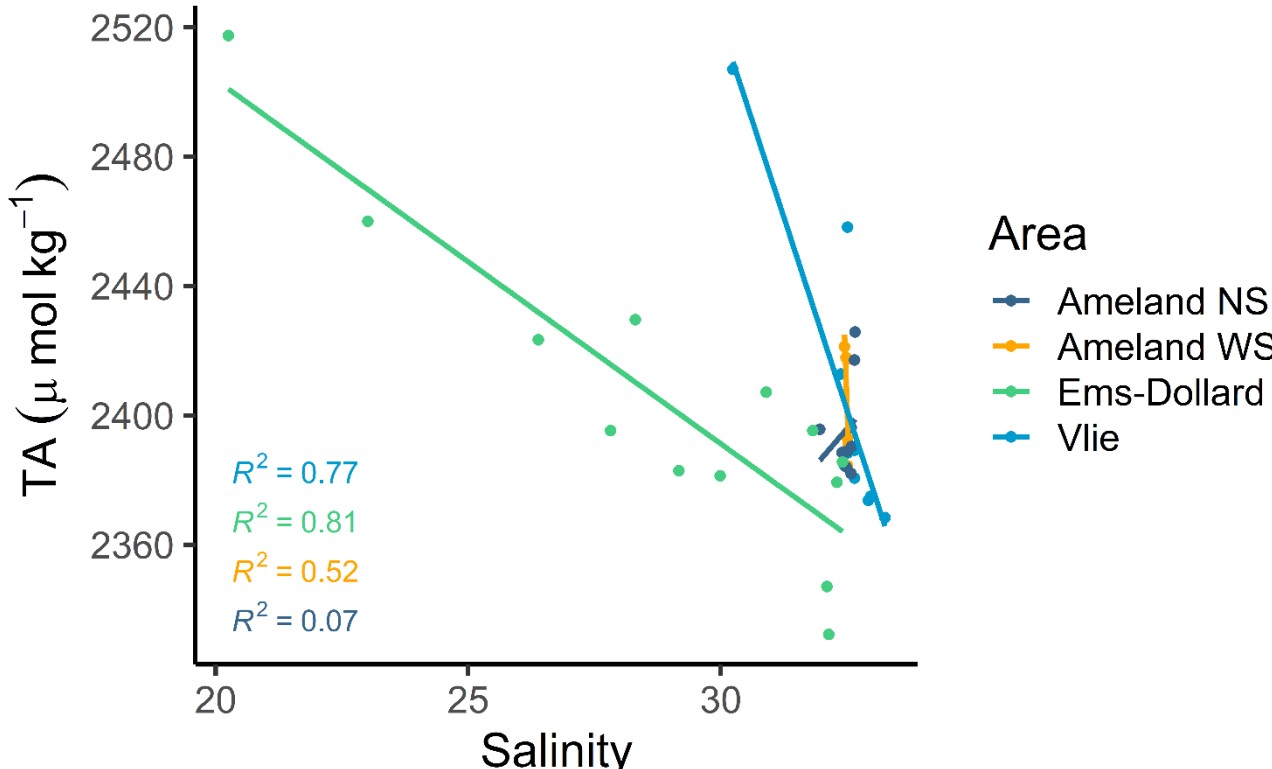


**Figure 3** Mixing plot of total alkalinity (TA) and salinity (PSU) in the North Sea side of Ameland and the Frisian Inlet
(Ameland NS), in the Wadden Sea site of Ameland (Ameland WS), around Schiermonnikoog and in the Ems-Dollard Inlet
(Ems-Dollard), and in the Vlie Inlet (Vlie).
**3.2 Tidal cycle**
We observed a half tidal cycle at an anchor station in the Ameland Inlet during ebb tide, to 1) identify potential TA sources
and 2) to quantify potential TA export to the North Sea. We identified patterns in several biogeochemical parameters in water
leaving the tidal flats (Fig. 4, Table B1). Temperature increased from 13.25 to 14.7 °C (Fig. 4a). Salinity was constant around
32.5 (Fig. 4b), which is in the range of coastal southern North Sea water excluding admixture of local fresh water sources.
During ebb tide, TA ranged from 2387 µmol TA kg$^{-1}$ during high tide to 2438 µmol TA kg$^{-1}$ during low tide (Fig. 4c). We
observed an increase of 51.6 µmol TA kg$^{-1}$ ($\Delta$TA) during ebb tide (6.8 h), resulting in a TA increase of 7.6 µmol TA kg$^{-1}$ h$^{-1}$
at the sampling location.
DIC concentrations behaved similar to TA with minimum values at high tide (2172 µmol DIC kg$^{-1}$), and maximum values
(2273 µmol DIC kg$^{-1}$) at low tide, resulting in an increase of 101.3 µmol DIC kg$^{-1}$ ($\Delta$DIC) or 14.9 µmol DIC kg$^{-1}$ h$^{-1}$ (Fig. 4d).
DIC increased almost twice as much as TA.
Nitrate increased during ebb tide by 0.92 µmol NO$_3^-$ L$^{-1}$ ($\Delta$NO$_3^-$) from a minimum of 1.26 µmol NO$_3^-$ L$^{-1}$ to a maximum of
2.17 µmol NO$_3^-$ L$^{-1}$ (Fig. 4e), resulting in a nitrate increase of 0.13 µmol NO$_3^-$ L$^{-1}$ h$^{-1}$.
Silicate showed a similar pattern with low values (1.8 µmol Si L$^{-1}$) at high tide increasing during ebb tide to a maximum of
11.2 µmol Si L$^{-1}$, resulting in a silicate increase ($\Delta$Si) of 9.4 µmol Si L$^{-1}$ or 1.4 µmol Si L$^{-1}$ h$^{-1}$ during ebb tide (Fig. 4f).
Ammonium increased from 3.47 µmol NH$_4^+$ L$^{-1}$ to 6.22 µmol NH$_4^+$ L$^{-1}$ during ebb tide (Fig. 4g), resulting in an ammonium
increase ($\Delta$NH$_4^+$) of 2.74 µmol NH$_4^+$ L$^{-1}$, or 0.4 µmol NH$_4^+$ L$^{-1}$ h$^{-1}$.
The calcite and aragonite saturation states had maximum values ($\Omega_{Ca}$ = 3.8, $\Omega_{Ar}$ = 2.4) at high tide and decreased to their
minimum ($\Omega_{Ca}$ = 3.1, $\Omega_{Ar}$ = 2.0) during ebb tide (Fig. 4h). The influence of the North Sea is indicated by the observed maximum
at high tide, which decreased during the ebb.
The seawater $p$CO$_2$ had minimum values at high tide (385.1 µatm) and increased up to 576.6 µatm during low tide (Fig. 4i).
Like $\Omega$, the maximum pH was 8.07 at high tide and decreased to a minimum (7.93) during ebb tide (Fig. 4j).
C$_{org}$:N ratios of SPM increased during ebb tide (Fig. 4k). A minimum C$_{org}$:N ratio of 5.6 was observed around high tide and
increased to a maximum of 13.0 during ebb tide. Simultaneously, the SPM concentration increased during ebb tide, from 12.8
mg SPM L$^{-1}$ to a maximum of 82.4 mg SPM L$^{-1}$ at the second last station (Table B1).

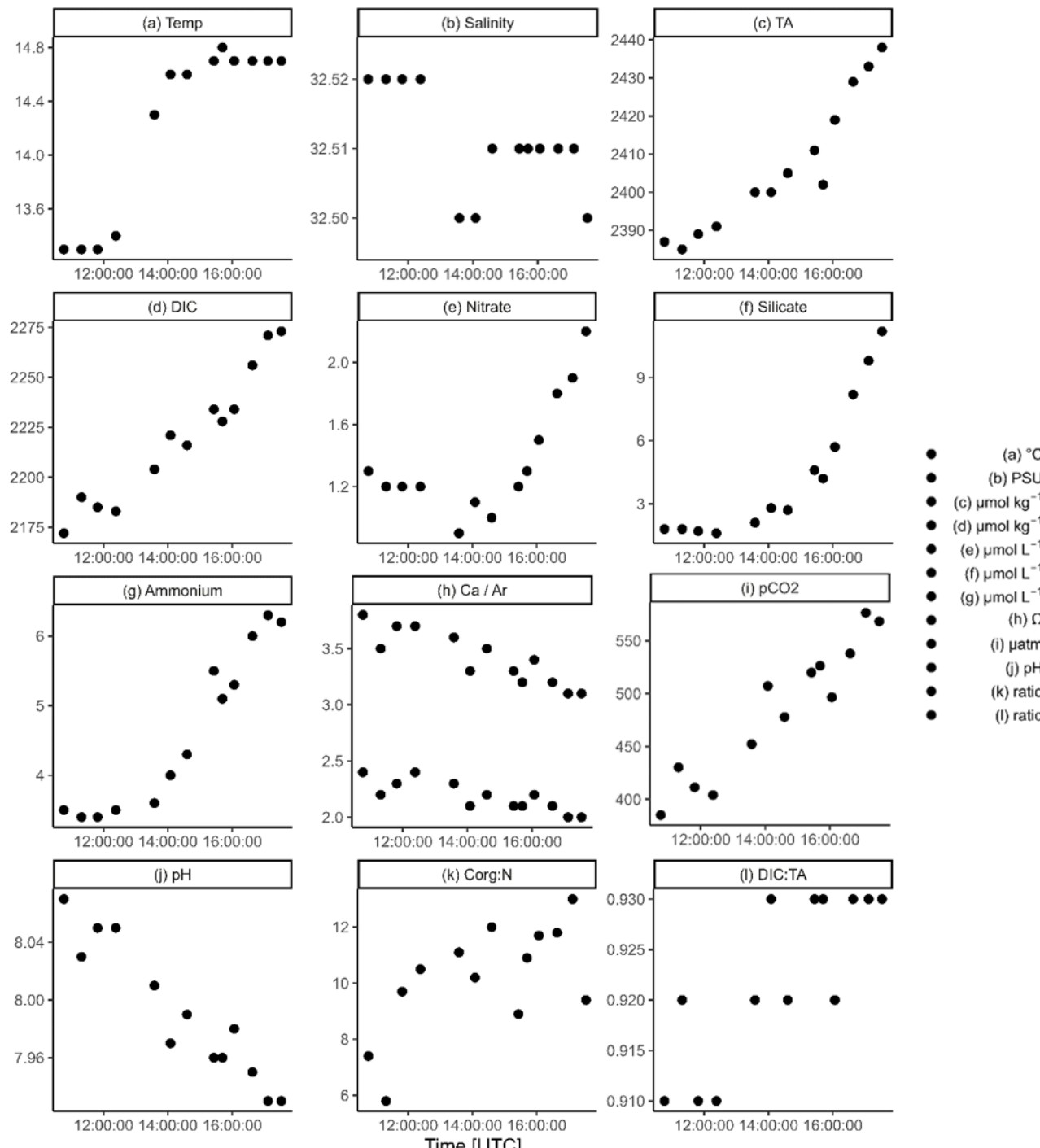


**Figure 4** A half tidal cycle from high tide to low tide. Temporal distribution of a) temperature, b) salinity, c) total alkalinity

(TA), d) dissolved inorganic carbon (DIC), e) nitrate, f) silicate, g) ammonium, h) calcite (upper points) and aragonite (lower

points) saturation states ($\Omega$), i) $pCO_2$ (µatm), j) pH, k) $C_{org}$:N ratio of SPM, and l) DIC:TA ratio. Note the different y-axes
and the +1 hour time difference between the local time and the UTC time.

## 3.3 TA generation

Tidal forcing leads to a bi-diurnal exchange between Wadden Sea and North Sea water. The tidal forcing also induces a strong
benthic-pelagic coupling (Huettel et al., 2003;Røy et al., 2008). Many studies support that the outflowing water exports
material from the sediment including remineralization products from organic matter (e.g., Billerbeck et al., 2006;Røy et al.,
2008). Here, we focus on the hypothesis that the sediments are a significant source of TA.
For a first rough estimate of a maximum TA export during ebb tide, we used the mean observed TA increase ($\Delta$TA / 2) of 25.8
µmol TA kg$^{-1}$ during ebb tide (in the Ameland Inlet, part of the Borndiep tidal basin), a tidal prism of 478 $*10^6$ m$^3$ of the
Borndiep tidal basin, and a share of intertidal flats of 53 % (Louters and Gerritsen, 1994). Assuming that only the intertidal
sediments exchange TA, we estimated a TA export of 6.6 Mmol TA per tide to the North Sea. Assuming two ebb tides and a
lunar cycle of 24.8 hours this would result in a daily export of 12.7 Mmol TA.
The significant correlation of TA and silicate ($R^2 = 0.93$), and the insignificant relation between TA and salinity ($R^2 = 0.32$),
as well as silicate and salinity ($R^2 = 0.21$), suggest that TA originates from the tidal flats in this part of the Dutch Wadden Sea
and is not from admixture carried by river runoff. The significant correlation between TA and silicate both during ebb tide
point to the same source (Fig. 5b).

To further elucidate potential TA sources in the Dutch Wadden Sea, we correlated TA with DIC, silicate, nitrate, and
ammonium in the half tidal cycle from high tide to low tide, respectively (Fig. 5).
The correlation between TA and DIC is a measure between anaerobic and aerobic processes. Our data show a strong positive
correlation between DIC and TA ($R^2 = 0.93$) with TA concentrations being higher than DIC concentrations (Fig. 5a). We
observed a release excess of DIC compared to TA as indicated by the slope of 1.89 and by an increase in DIC ($\Delta$DIC = 101.3
µmol kg$^{-1}$) almost twice as high as TA ($\Delta$TA = 51.6 µmol kg$^{-1}$) (Fig. 5a). This excess DIC may be caused by strong $CO_2$
production due to high aerobic OM degradation, which can be supported by seawater being supersaturated in $pCO_2$ with respect
to the atmosphere (Fig. 4i). The TA increase can be fueled by various processes which we will discuss below. We detected a
linear positive correlation of increasing TA and silicate ($R^2 = 0.93$) during ebb tide, supporting pore water outflow (Fig. 5b)
as pore water is the major Si source during summer (van Bennekom et al., 1974). A stronger influence of the pore water with
ongoing ebb tide is indicated by increasing values. The positive correlation between nitrate and TA ($R^2 = 0.67$) (Fig. 5c) was
less strong than the correlations between TA and DIC, and TA and Si, which could be traced back to an effect of the first four
sampling points that were probably at the tipping point from high tide to low tide. In the remaining samples, the increasing
nitrate and TA concentrations suggest a stronger TA generation than nitrate production, balancing TA that may be consumed
by nitrification (i.e., nitrate production).

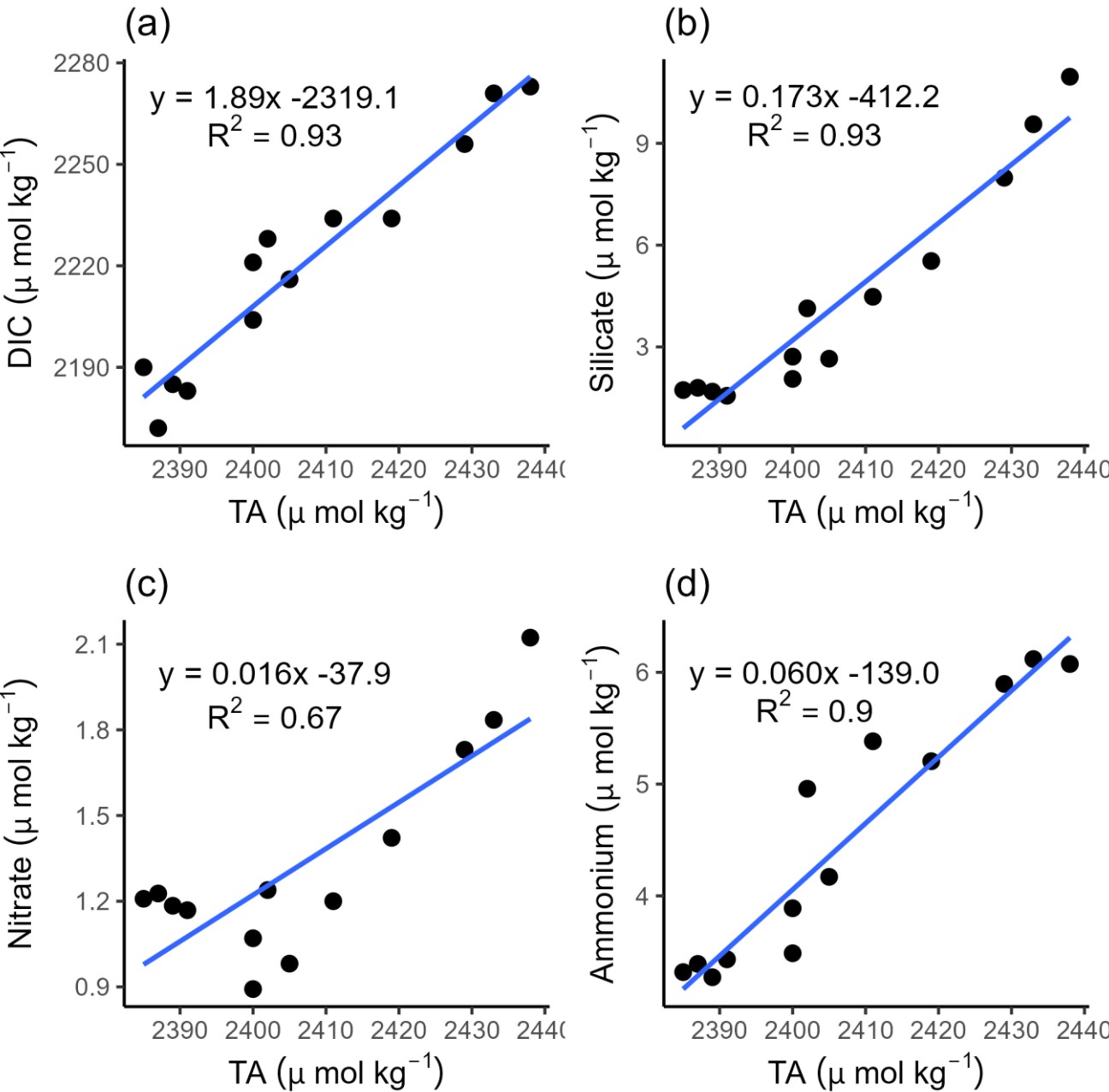


**Figure 5** Correlations of TA with a) dissolved inorganic carbon (DIC), b) silicate, c) nitrate, and d) ammonium during ebb
tide in the Ameland Inlet.

## 4 Discussion

### 4.1 Spatial TA variability

Hoppema (1990) reported TA distributions in the westernmost part of the Dutch Wadden Sea around the barrier islands Texel, Vlieland, and Terschelling. He focused on the tidal basins drained by the tidal inlets Marsdiep and Vlie located more to the west than our sampling stations (not visible on the map). Hoppema (1990) did not observe a continuous increase of salinity in the Wadden Sea from the fresh water source towards the North Sea and associated this to the influence of tidal differences and an arbitrary sampling scheme. The presence (dominance) of North Sea water in the Dutch Wadden Sea and on the tidal flats is supported by our transect data, which show relatively high salinities at coastal North Sea level. Brackish salinities were only detected in the Ems-Dollard Inlet, which receives fresh water from the river Ems, and close to Harlingen and Lauwersoog, which have direct fresh water inflows by smaller rivers and streams. The absence of clear salinity gradients in the more eastern part of the Dutch Wadden Sea investigated in our study suggest that most of the IJsselmeer discharge was exchanged with the North Sea through the Marsdiep (e.g., Duran-Matute et al., 2014).

The spatial TA data by Hoppema (1990), show lower TA concentrations at stations with more fresh water influence and higher TA concentrations in the tidal inlets. The data of this study also show high TA concentrations in the tidal inlets, suggesting TA generation in sediments, which is possibly fueled by high imports of nutrients and OM (van Beusekom and De Jonge, 2002). The even higher TA concentrations at stations with lower salinities close to the mainland observed in this study also show the influence from the catchment area on the coast, and possibly TA generation in the shallow sediments near the coast. In May 1986, Hoppema (1990) found TA concentrations ranging between 2319 and 2444 µmol TA kg$^{-1}$ at salinities between 18.62 and 29.17. Our lowest observed TA concentration was 2332 µmol TA kg$^{-1}$ at a salinity of 32.14, and our highest TA concentration was 2517 µmol TA kg$^{-1}$ at a salinity of 20.25 close to the coastal mainland. A comparison of both studies shows that the general TA levels are in a similar range, but that the spatial gradients are opposite.

A conservative mixing between TA and salinity is only visible in the Ems-Dollard Inlet and the Vlie Inlet (Fig. 3). While the conservative mixing in the Ems-Dollard Inlet is more dominated by the fresh water discharge from the Ems River, the conservative mixing in the Vlie Inlet is more dominated by North Sea water passing through this deep inlet and allowing more North Sea water to be transported towards the coast. After the Marsdiep Inlet, the Vlie Inlet has the highest average tidal prism and is the second largest inlet in the Dutch Wadden Sea (Elias et al., 2012). Similar to our findings, Hoppema (1990) noted a linear mixing of TA and salinity in the Vlie Inlet, and suspected a lower fresh water contribution there as well, which is in accordance with model data (Duran-Matute et al., 2014).

In the Ems-Dollard Inlet, conservative mixing was observed, indicating minor contributions from other sources. In a previous study, Norbisrath et al. (2023) observed very high TA concentrations and TA generation in the upper tidal river of the highly turbid Ems Estuary, which may explain the high levels of TA in the Ems-Dollard Inlet (at low salinities) observed in this study.

Hoppema (1990) also observed a range of TA concentrations in the Dutch Wadden Sea and related these to different sinks and
sources. TA sinks can be calcium carbonate ($CaCO_3$) precipitation, or extraction of seawater carbonate by mollusks (e.g., Chen
and Wang, 1999;Hoppema, 1990). Variable fresh water inflows can either serve as a sink or a source (e.g., Chen and Wang,
1999;Hoppema, 1990). Other TA sources can be $CaCO_3$ dissolution, anaerobic metabolic processes in the sediment, or erosion
of TA enhancing sediments (e.g., Hoppema, 1990;Chen and Wang, 1999).
Except for the Ems-Dollard Inlet and close to Harlingen, we observed mainly marine salinities (> 30) but higher TA values in
the Dutch Wadden Sea than in the North Sea. We therefore exclude possible TA sinks and focus only on TA sources. According
to Hoppema (1990), the main causes for TA variations in the Dutch Wadden Sea were fresh water inflows and sources in the
sediment. In our study, fresh water inflows with high TA concentrations were only observed in the Ems-Dollard Inlet, but not
around the islands and the tidal flats. For a further TA source identification in the Dutch Wadden Sea, we investigated the TA
variability during ebb tide in a tidal channel close to Ameland.

### 279  4.2 Determination of TA generation

Burt et al. (2016) and Schwichtenberg et al. (2020) indicated TA generation in the Wadden Sea as an important source for the
North Sea's carbon storage capacity. Here, we want to further identify TA generation and potential TA sources.
In a study from the late 1980s, Hoppema (1993) observed a tidal cycle in the Marsdiep in May and September. Focusing on
TA, DIC, and oxygen, he also observed increasing TA values during ebb tide and assumed the tidal flats and discharging rivers
and canals as TA sources. Comparing our present TA data and the historical TA data, there is not a large difference in the
range of values observed during a tidal cycle. However, a further in-depth interpretation and comparison of both TA data sets
is limited by the low number of data, leading us to focus on TA generation during our cruise.
We made a very rough first estimate of the daily TA export. By using a 3D ecosystem model, Schwichtenberg et al. (2020)
estimated an annual export of 10 to 14 Gmol TA $yr^{-1}$ for the entire Dutch Wadden Sea. Given that the Borndiep tidal basin
covers about 14% of the Dutch Wadden Sea and assuming no seasonal dynamics, our estimate of 12.7 Mmol $d^{-1}$ compares
well with the annual averaged model estimate of 4.6 Mmol TA $d^{-1}$, but the overestimation suggests that seasonal dynamics
may be involved. Since our TA export based only on a half tidal observation, the inclusion of it into the model used by
Schwichtenberg et al. (2020) would be unreliable (pers. comm. J. Pätsch, 2022). To test whether the observed TA generation
matches their suggested TA export, observational data of at least each season are required to run the model and gain a
representative result (pers. comm. J. Pätsch, 2022).

### 295  4.3 TA source attribution

### 296  4.3.1 Local sediment outwash

In order to gain further insight into potential sources of TA, we compared our TA and nutrient data. The main focus was on
dissolved silicate (Si) as van Bennekom et al. (1974) showed that this nutrient is depleted in the Wadden Sea during the spring
diatom blooms and further showed that pore water is the main source of dissolved silicate during summer. It is important to
note that winter concentrations in the Rhine (main contributor to the IJsselmeer) have not changed much since the 1970s and
showing maximum concentrations of about 125 µmol Si L$^{-1}$ in winter and clear seasonal dynamics due to uptake by diatoms
(unpublished results based on data provided by Pätsch (2024); available through https://wiki.cen.uni-
hamburg.de/ifm/ECOHAM/DATA_RIVER). We identified a silicate increase of 1.4 µmol Si L$^{-1}$ h$^{-1}$ during ebb tide. Due to
the absence of large estuaries nearby and salinity consistently being above 32 at our tidal sampling station near the island of
Ameland, we exclude fresh water runoff as a major silicate source and indicate TA sources within the Wadden Sea.
Submarine groundwater discharge (SGD) was identified as a source for nutrient fluxes in tidal flat ecosystems in previous
studies (e.g., Billerbeck et al., 2006;Røy et al., 2008;Santos et al., 2021;Waska and Kim, 2011;Wu et al., 2013). Since we
observed relatively constant marine salinities, we suspect that deep pore water flow (e.g., Røy et al., 2008) enriched with
nutrients act as a source for our observed increasing TA and nutrients parameters. TA generation in tidal flats was also observed
by Faber et al. (2014), who focused on a large macro tidal embayment in southern Australia. They also found increasing TA
values during ebb tide, associated the TA increase with a higher fraction of pore water, and determined the tidal cycle as the
controlling force for pore water exchange. Their findings and the observed silicate outwash support our assumption that TA is
generated in the sediments of the tidal flats and is washed out during ebb tide. In addition, we exclude lateral advected signals
from more western regions as the Vlie Inlet, since the TA concentrations in the surface transect samples in the Vlie Inlet
(except of the two samples close to the coastal mainland near Harlingen) were in the same range as the other observed TA
concentrations and were smaller than the increasing TA concentrations during ebb tide. Both increases in TA and silicate are
tidal signals, and we identify TA generation in the sediments of the tidal flats here as the major local TA source.
**4.3.2 TA generating processes**
The observed TA increase of 7.6 µmol TA kg$^{-1}$ h$^{-1}$ and the silicate increase of 1.4 µmol Si L$^{-1}$ h$^{-1}$ indicated an excess of TA
compared to silicate (also Fig. 5b). Considering a supposed TA:Si ratio of 2:1 (Marx et al. 2017), the observed 1.4 µmol Si L$^{-1}$
h$^{-1}$ would then account for a TA generation of 2.8 µmol TA kg$^{-1}$ h$^{-1}$. High silicate concentrations in tidal flat pore water (Rutgers
van der Loeff, 1974) and in situ production of silicate from dissolving diatom frustules are the most probable sources of the
silicate (e.g., van Bennekom et al., 1974). Since we observed more TA generated than silicate being washed out, other
biogeochemical processes must be responsible for the TA generation in the sediments of the tidal flats in the Dutch Wadden
Sea.

We exclude $CaCO_3$ dissolution as TA source in the water column, since the $\Omega$ values were supersaturated with $\Omega > 1$ (Fig. 4h,
Table B1). The continuous calcite supersaturation nicely indicated the inflow and dominance of North Sea water during the
flood, with $\Omega$ values similar to previously observed North Sea values ($\Omega \sim 3.5$ to 4) (Charalampopoulou et al., 2011;Carter et
al., 2014). In pore water, carbonate undersaturation and associated $CaCO_3$ dissolution can only be driven metabolically, due
to $CO_2$ production by OM remineralization, or due to the reoxidation of compounds reduced previously by anaerobic processes
(Brenner et al., 2016;Jahnke et al., 1994).

Other potential sources of TA generation in the sediments can be further narrowed down by a more detailed interpretation of
changes in DIC ($\Delta$DIC) and TA ($\Delta$TA) during ebb tide, and their combination with various nutrient ratios. The correlation of
DIC and TA reveals an increase in DIC ($\Delta$DIC) almost twice as high as in TA ($\Delta$TA) (Fig. 5a), as indicated by the slope of
1.89. The high $\Delta$DIC points to high aerobic OM degradation and remineralization, resulting in high $CO_2$ production, which is
also indicated by seawater being supersaturated in $p$CO$_2$. High aerobic OM degradation was also previously observed in the
heterotrophic Wadden Sea (e.g., De Beer et al., 2005;van Beusekom et al., 1999), assuming an OM degradation and
remineralization occurring in the water and sediment in about equal parts (van Beusekom et al., 1999). High OM degradation
is indicated by the increasing $C_{org}$:N ratios of SPM during ebb tide (Fig. 4k, Table B1). Because we observed constant coastal
North Sea salinities, we rule out fresh water runoff and terrestrial signals as source for the increasing $C_{org}$:N ratios of SPM.
We assume that fresh OM is rapidly degraded in the water column, and the older OM settles on and in the sediment where the
degradation continues and where it is resuspended at the low prevailing water levels during ebb. Therefore, we assume that
the increase of SPM concentrations and their $C_{org}$:N ratios is an indicator for older and more refractory OM. The increase in
TA concentrations point to anaerobic processes, $CaCO_3$ dissolution, or a combination thereof as TA sources occurring in the
sediments.

For an upper bound estimate of sedimentary $CaCO_3$ dissolution as source of TA, we considered a $\Delta$DIC: $\Delta$TA ratio of 1:2.
Considering this ratio and the observed $\Delta$TA of 51.6 µmol TA kg$^{-1}$, $CaCO_3$ dissolution would lead to a $\Delta$DIC of 25.8 µmol
DIC kg$^{-1}$. The remaining 75.5 µmol DIC kg$^{-1}$ (101.3 – 25.8 µmol DIC kg$^{-1}$) of the observed $\Delta$DIC in this study could then be
produced by OM degradation and remineralization, and would, using the theoretical expected Redfield ratio of C:N (6.6) for
fresh OM (Hickel, 1984), correspond to an estimated dissolved inorganic nitrogen (DIN) production of 11.4 µmol DIN kg$^{-1}$.
However, this estimated DIN production (11.4 µmol DIN kg$^{-1}$) of OM degradation and remineralization exceeds the observed
increase of $\Delta$DIN (3.97 µmol DIN L$^{-1}$; Table B1, sum of $NO_3^-$, $NO_2^-$ and $NH_4^+$) during ebb tide. Based on this estimation and
the assumption that all DIN produced is released and thus lost, TA is probably produced by $CaCO_3$ dissolution and anaerobic
metabolic processes other than denitrification in the sediment. In addition to that, and with an N-focused perspective, the DIN
loss also hints to the occurrence of other processes that consume nitrogen species but have no net effect on TA, such as
anammox and coupled nitrification-denitrification (Hu and Cai, 2011;Middelburg et al., 2020). The suggested DIN loss can
be supported by considering the marine DIN:Si ratio, which is supposed to be about 1:1 (Brzezinski, 1985). We observed
DIN:Si ratios decreasing from 2.7 to 0.8 during ebb tide, showing that both parameter concentrations increased, whereby DIN
concentrations increased less than silicate concentrations. The silicate excess with respect to DIN at the end of ebb tide supports
the DIN loss.

Denitrification, the anaerobic irreversible reduction of $NO_3^-$ to $N_2$ that generates 0.9 mole TA by using 1 mole $NO_3^-$ as electron acceptor (Chen and Wang, 1999) is a net TA source. Denitrification depends on the supply of nitrate, which seasonally varies (van der Zee and Chou, 2005 and references therein). Generally, nitrate is depleted in summer due to high photosynthetic activity and occurs in higher concentrations in winter (Kieskamp et al., 1991;Jensen et al., 1996;van der Zee and Chou, 2005). This seasonality leads to denitrification rates also being lower in summer and higher in winter (Kieskamp et al., 1991;Jensen et al., 1996). In previous studies, Faber et al. (2014) identified denitrification as a minor source of TA due to low denitrification rates, and also Kieskamp et al. (1991) observed low denitrification rates in the Wadden Sea, with low nitrate concentrations ($< 2.5$ µmol $L^{-1}$) in the water column. We observed nitrate concentrations ($< 2.17$ µmol $L^{-1}$) lower than the concentration sufficient for denitrification assumed by Kieskamp et al. (1991). Therefore, we do not exclude denitrification, but suspect it as a minor source of TA in the Dutch Wadden Sea at least in spring and summer due to the seasonal lack of nitrate. Thomas et al. (2009) detected TA seasonality in the southern bight of the North Sea, which is also influenced by the TA generation in the Wadden Sea. In addition, the estimated DIN production compared to the observed DIN not only hints to other N consuming processes that have no effect on TA, but also suggests that allochthonous nitrate would be needed to fuel the TA increase by denitrification.

The simultaneous increase of ammonium and TA (Fig. 4c, 4d, 5d) is important to notice, because under oxic conditions the occurrence of ammonium is coupled with nitrification, a process that consumes ammonium and also TA (Chen and Wang, 1999). However, under anoxic conditions, such as in deeper sediment layers, ammonium cannot be reoxidized, accumulates, and is washed out during ebb tide. Since we observed low nitrate concentrations and rule out terrestrial nitrate inputs here, the increase in ammonium and TA implies the occurrence of other anaerobic processes of the redox system, such as sulfate and iron reduction, to generate TA in the deeper, anoxic sediment layers in the Dutch Wadden Sea.

Sulfate reduction followed by iron reduction and the formation and burial of pyrite are net sources of TA, since TA consumption by reoxidation is excluded when buried in sediments (Berner et al., 1970;Faber et al., 2014). Whether and to what extent these processes contribute to TA generation in the deeper sediments of the Dutch Wadden Sea cannot be further identified without the necessary data. However, sulfate reduction was also mentioned as source of TA by Thomas et al. (2009), and the temporary slight appearance of noticeable sulfuric odor could be another indirect indicator for the occurrence of sulfate reduction. In previous studies of tidal flats in the German Wadden Sea, Beck et al. (2008a);(2008b) observed increasing TA concentrations with depth and identified sulfate reduction as the most important process for anaerobic OM remineralization in pore water cores.

A strict comparison of the more northern (north of the Elbe Estuary) and the more western (Texel – Elbe Estuary) parts of the Wadden Sea is difficult because the areas vary in terms of OM import and eutrophication effects (van Beusekom et al., 2019), sediment composition, and extent between the barrier islands and the mainland, all of which influence the occurrence and interaction of biogeochemical processes (Schwichtenberg et al., 2020). The area characteristics of the northern and western

Wadden Sea differ especially in terms of OM turnover being lower in the norther Wadden Sea. However, a previous study by
Brasse et al. (1999) identified high TA and DIC concentrations in the sediment of the North Frisian Wadden Sea and identified
$CaCO_3$ dissolution and sulfate reduction as major TA sources, which is consistent with our findings.

## 5 Conclusion

The Dutch Wadden Sea is a unique and highly dynamic ecosystem. We observed higher TA values in the Dutch Wadden Sea
than in the North Sea and identified the Dutch Wadden Sea as a TA source for the North Sea's carbonate system. Compared
to previous studies (Hoppema, 1990, 1993), the TA values we observed were in a similar range, with high TA values in the
tidal basins. Beside the need for seasonal observations, future work should also focus on regional and seasonal impacts of fresh
water inflows of TA on the TA status in the Dutch Wadden Sea.
By observing salinity and using dissolved silicate as a tracer, we excluded fresh water and river runoff as significant TA sources
on the tidal flats, and instead, deduced local outwash from the sediments as sources of TA. Considering various stoichiometries,
we suggest that $CaCO_3$ dissolution generates TA in the more upper oxic sediment layers, and anaerobic, metabolic processes
such as denitrification, sulfate and iron reduction are potential TA sources in the deeper anoxic sediment layers. However, in
spring and early summer, denitrification seems to play a minor role in generating TA in the sediments of the Dutch Wadden
Sea due to seasonality and associated limited nitrate availability.
**6 Appendices**
**Appendix A**

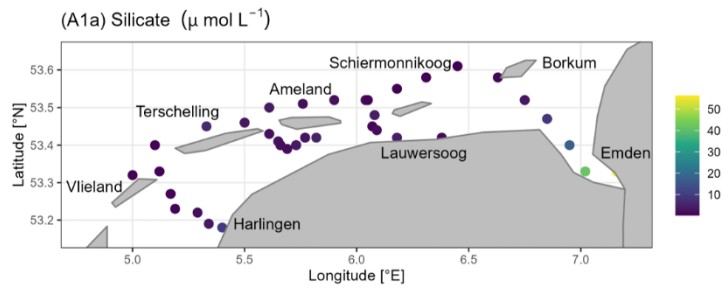

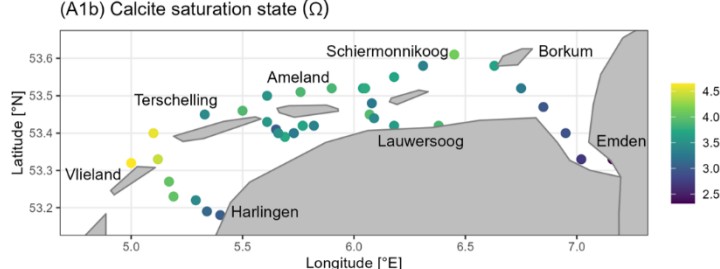

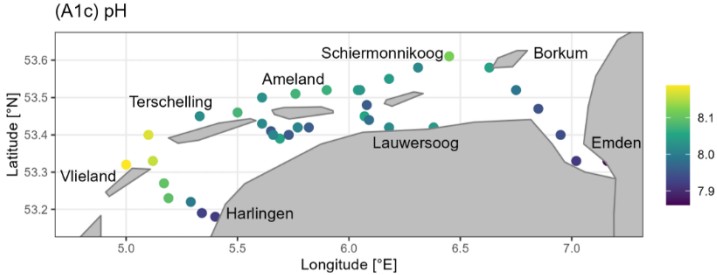

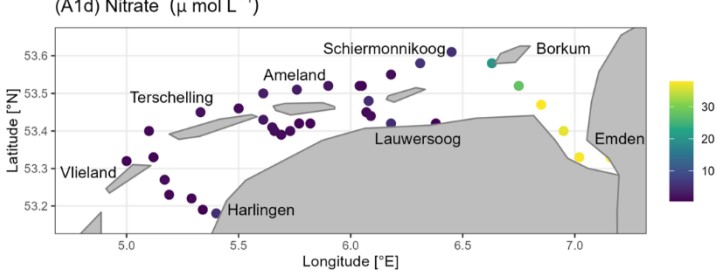

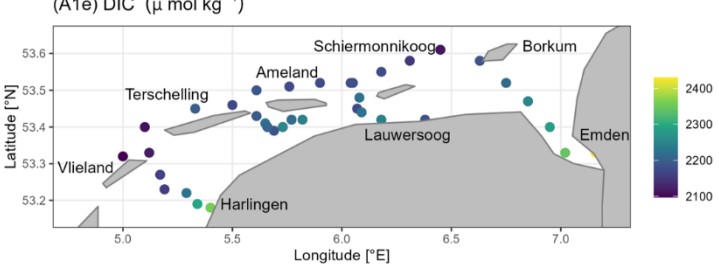


**Figure A1** Spatial distribution of A1a) silicate (Si; $\mu$mol L$^{-1}$), A1b) calcite saturation state ($\Omega$), A1c) pH, A1d) nitrate (NO$_3^-$;
$\mu$mol L$^{-1}$), and A1e) dissolved inorganic carbon (DIC; $\mu$mol kg$^{-1}$) from surface water samples in May 2019.

**Appendix B**

**Table B1** Half tidal cycle sample parameter during ebb tide. Sample no. 545 is the first sample at high tide and sample no. 557 is the last sample at low tide on 21 May 2019 (53.38°N & 5.62°E). Shown are values of temperature (Temp), salinity (Sal), total alkalinity (TA), dissolved inorganic carbon (DIC), silicate (Si), nitrate ($NO_3^-$), nitrite ($NO_2^-$), ammonium ($NH_4^+$), phosphate ($PO_4^{3-}$), dissolved inorganic nitrogen (DIN), the amount of carbon (C) and organic carbon ($C_{org}$) of SPM, the amount of nitrogen (N) of SPM, the calcite (Ca) and aragonite (Ar) saturation states, the pH, and the seawater partial pressure of $CO_2$ ($pCO_2$) per sample.

| Sample No. | Time [UTC] | Temp [°C] | Sal [PSU] | TA / DIC [$\mu$mol kg$^{-1}$] | Si [$\mu$mol L$^{-1}$] | $NO_3^-$ [$\mu$mol L$^{-1}$] | $NO_2^-$ [$\mu$mol L$^{-1}$] | $NH_4^+$ [$\mu$mol L$^{-1}$] | $PO_4^{3-}$ [$\mu$mol L$^{-1}$] |
|---|---|---|---|---|---|---|---|---|---|
| 545 | 10:46 | 13.26 | 32.52 | 2387 / 2172 | 1.84 | 1.26 | 0.19 | 3.47 | 0.12 |
| 546 | 11:19 | 13.25 | 32.52 | 2385 / 2190 | 1.77 | 1.24 | 0.19 | 3.40 | 0.11 |
| 547 | 11:49 | 13.28 | 32.52 | 2389 / 2185 | 1.72 | 1.21 | 0.19 | 3.35 | 0.11 |
| 548 | 12:23 | 13.38 | 32.52 | 2391 / 2183 | 1.6 | 1.19 | 0.19 | 3.52 | 0.12 |
| 549 | 13:35 | 14.32 | 32.50 | 2400 / 2204 | 2.11 | 0.91 | 0.25 | 3.57 | 0.32 |
| 550 | 14:05 | 14.61 | 32.50 | 2400 / 2221 | 2.78 | 1.09 | 0.29 | 3.98 | 0.42 |
| 551 | 14:36 | 14.64 | 32.51 | 2405 / 2216 | 2.72 | 1.01 | 0.29 | 4.27 | 0.47 |
| 552 | 15:26 | 14.73 | 32.51 | 2411 / 2234 | 4.59 | 1.23 | 0.34 | 5.51 | 0.57 |
| 553 | 15:42 | 14.77 | 32.51 | 2402 / 2228 | 4.24 | 1.26 | 0.33 | 5.08 | 0.54 |
| 554 | 16:04 | 14.72 | 32.51 | 2419 / 2234 | 5.66 | 1.46 | 0.36 | 5.33 | 0.54 |
| 555 | 16:38 | 14.66 | 32.51 | 2428 / 2256 | 8.18 | 1.77 | 0.43 | 6.04 | 0.58 |
| 556 | 17:07 | 14.68 | 32.51 | 2433 / 2271 | 9.79 | 1.87 | 0.47 | 6.27 | 0.62 |
| 557 | 17:32 | 14.70 | 32.50 | 2438 / 2273 | 11.22 | 2.17 | 0.50 | 6.22 | 0.63 |

| Sample No. | Time [UTC] | DIN [$\mu$mol L$^{-1}$] | C / $C_{org}$ (SPM) [$\mu$mol L$^{-1}$] | N (SPM) [$\mu$mol L$^{-1}$] | $C_{org}$:N (SPM) | SPM [mg L$^{-1}$] | Ca / Ar [$\Omega$] | pH | $pCO_2$ [$\mu$atm] |
|---|---|---|---|---|---|---|---|---|---|
| 545 | 10:46 | 4.93 | 86.8 / 65.1 | 8.8 | 7.4 | 12.8 | 3.8 / 2.4 | 8.07 | 385.1 |
| 546 | 11:19 | 4.83 | 72.7 / 42.4 | 7.4 | 5.8 | 8.7 | 3.5 / 2.3 | 8.03 | 430.2 |
| 547 | 11:49 | 4.76 | 112.4 / 93.4 | 9.6 | 9.7 | 15.4 | 3.7 / 2.3 | 8.05 | 411.4 |
| 548 | 12:23 | 4.91 | 108.5 / 104.6 | 9.9 | 10.5 | 16.8 | 3.7 / 2.4 | 8.05 | 404.1 |
| 549 | 13:35 | 4.73 | 111.1 / 97.8 | 8.8 | 11.1 | 13.9 | 3.6 / 2.3 | 8.01 | 452.3 |
| 550 | 14:05 | 5.37 | 233.0 / 180.3 | 17.7 | 10.2 | 32.2 | 3.3 / 2.1 | 7.97 | 507.2 |
| 551 | 14:36 | 5.56 | 193.2 / 174.3 | 14.5 | 12.0 | 29.6 | 3.5 / 2.2 | 7.99 | 477.9 |
| 552 | 15:26 | 7.08 | 248.6 / 163.5 | 18.4 | 8.9 | 34.3 | 3.3 / 2.1 | 7.96 | 520.0 |
| 553 | 15:42 | 6.67 | 257.6 / 199.3 | 18.3 | 10.9 | 41.6 | 3.2 / 2.1 | 7.95 | 526.4 |
| 554 | 16:04 | 7.15 | 324.4 / 271.1 | 23.2 | 11.7 | 55.0 | 3.4 / 2.2 | 7.98 | 496.6 |
| 555 | 16:38 | 8.24 | 440.4 / 345.2 | 29.2 | 11.8 | 75.7 | 3.2 / 2.1 | 7.95 | 538.0 |
| 556 | 17:07 | 8.61 | 430.5 / 363.3 | 27.9 | 13.0 | 82.4 | 3.1/ 2.0 | 7.93. | 576.6 |

| 557 | 17:32 | 8.90 | 308.9 / 199.1 | 21.2 | 9.4 | 48.8 | 3.1 / 2.0 | 7.93 | 568.4 |



**Table B2** Half tidal cycle sample parameter during high tide for comparison. Sample no. 564 is the first sample at low tide
and sample no. 578 is the last sample at high tide on 23 May 2019 (53.39°N & 5.63°E, 5.62°E*). Shown are values of
temperature (Temp), salinity (Sal), total alkalinity (TA), dissolved inorganic carbon (DIC), silicate (Si), nitrate ($NO_3^-$), nitrite
($NO_2^-$), ammonium ($NH_4^+$), phosphate ($PO_4^{3-}$), dissolved inorganic nitrogen (DIN), the amount of carbon (C) and organic
carbon ($C_{org}$) of SPM, the amount of nitrogen (N) of SPM, the calcite (Ca) and aragonite (Ar) saturation states, the pH, and the
seawater partial pressure of $CO_2$ ($pCO_2$) per sample.

| Sample No. | Time [UTC] | Temp [°C] | Sal [PSU] | TA / DIC [μmol kg$^{-1}$] | Si [μmol L$^{-1}$] | $NO_3^-$ [μmol L$^{-1}$] | $NO_2^-$ [μmol L$^{-1}$] | $NH_4^+$ [μmol L$^{-1}$] | $PO_4^{3-}$ [μmol L$^{-1}$] |
|---|---|---|---|---|---|---|---|---|---|
| 564 | 05:09 | 14.04 | 32.66 | 2431 / 2246 | 8.53 | 1.25 | 0.47 | 3.31 | 0.38 |
| 565 | 05:32 | 14.02 | 32.68 | 2441 / 2287 | 9.14 | 1.26 | 0.45 | 3.08 | 0.37 |
| 566 | 06:01 | 13.95 | 32.69 | 2436 / 2284 | 8.88 | 1.33 | 0.38 | 2.46 | 0.34 |
| 567 | 06:33 | 14.16 | 32.69 | 2443 / 2284 | 8.68 | 0.95 | 0.37 | 2.37 | 0.33 |
| 568 | 07:02 | 14.21 | 32.69 | 2432 / 2280 | 6.94 | 0.75 | 0.34 | 2.63 | 0.32 |
| 569 | 07:31 | 14.15 | 32.55 | 2401 / 2223 | 2.12 | 0.98 | 0.27 | 4.12 | 0.33 |
| 570 | 08:04 | 14.20 | 32.55 | 2403 / 2218 | 2.10 | 1.04 | 0.27 | 3.88 | 0.30 |
| 571 | 08:35 | 14.27 | 32.55 | 2409 / 2228 | 2.15 | 0.92 | 0.25 | 4.18 | 0.32 |
| 572 | 09:04 | 14.37 | 32.53 | 2400 / 2209 | 1.88 | 1.00 | 0.22 | 3.86 | 0.26 |
| 573 | 09:34 | 14.16 | 32.52 | 2398 / 2200 | 1.70 | 1.03 | 0.21 | 3.51 | 0.21 |
| 574* | 10:02 | 14.17 | 32.52 | 2391 / 2197 | 1.72 | 1.07 | 0.21 | 3.40 | 0.18 |
| 575* | 10:34 | 14.11 | 32.51 | 2389 / 2195 | 1.78 | 1.18 | 0.20 | 3.45 | 0.16 |
| 576 | 11:04 | 14.21 | 32.50 | 2390 / 2187 | 1.76 | 1.12 | 0.19 | 3.29 | 0.14 |
| 577 | 11:34 | 14.50 | 32.51 | 2399 / 2193 | 1.66 | 1.10 | 0.20 | 3.32 | 0.16 |
| 578 | 12:03 | 13.96 | 32.51 | 2390 / 2187 | 1.75 | 1.41 | 0.19 | 3.72 | 0.11 |

| Sample No. | Time [UTC] | DIN [μmol L$^{-1}$] | C / $C_{org}$ (SPM) [μmol L$^{-1}$] | N (SPM) [μmol L$^{-1}$] | $C_{org}$:N (SPM) | SPM [mg L$^{-1}$] | Ca / Ar [Ω] | pH | $pCO_2$ [μatm] |
|---|---|---|---|---|---|---|---|---|---|
| 564 | 05:09 | 5.03 | 353.7 / 253.2 | 27.5 | 9.2 | 52.3 | 3.0 / 2.2 | 7.99 | 490.3 |
| 565 | 05:32 | 4.78 | 333.5 / 220.1 | 26.1 | 8.4 | 49.7 | 3.0 / 1.9 | 7.92 | 592.9 |
| 566 | 06:01 | 4.17 | 330.3 / 232.9 | 25.5 | 9.1 | 51.7 | 2.9 / 1.9 | 7.91 | 600.3 |
| 567 | 06:33 | 3.68 | 274.7 / 195.7 | 21.8 | 9.0 | 36.9 | 3.0 / 1.9 | 7.92 | 582.6 |
| 568 | 07:02 | 3.72 | 317.8 / 220.2 | 24.5 | 9.0 | 46.1 | 2.9 / 1.9 | 7.91 | 601.8 |
| 569 | 07:31 | 5.37 | 88.6 / 59.1 | 7.0 | 8.5 | 14.7 | 3.3 / 2.1 | 7.98 | 500.7 |
| 570 | 08:04 | 5.20 | 96.8 / 73.6 | 8.8 | 8.4 | 18.1 | 3.4 / 2.2 | 7.99 | 482.6 |
| 571 | 08:35 | 5.35 | 114.2 / 109.6 | 9.9 | 11.0 | 14.8 | 3.3 / 2.1 | 7.98 | 497.6 |

| 572 | 09:04 | 5.08 | 107.5 / 73.9 | 9.9 | 7.5 | 16.4 | 3.5 / 2.2 | 8.00 | 466.6 |
|---|---|---|---|---|---|---|---|---|---|
| 573 | 09:34 | 4.75 | 82.1 / 72.7 | 7.2 | 10.0 | 11.8 | 3.6 / 2.3 | 8.02 | 445.3 |
| 574* | 10:02 | 4.68 | 85.2 / 62.9 | 7.2 | 8.7 | 9.9 | 3.5 / 2.3 | 8.01 | 450.5 |
| 575* | 10:34 | 4.83 | 83.5 / 65.9 | 7.2 | 9.2 | 11.1 | 3.5 / 2.3 | 8.01 | 449.6 |
| 576 | 11:04 | 4.60 | 82.7 / 52.1 | 8.2 | 6.3 | 8.5 | 3.7 / 2.3 | 8.03 | 429.9 |
| 577 | 11:34 | 4.62 | 65.8 / 50.8 | 6.5 | 7.8 | 7.2 | 3.7 / 2.4 | 8.03 | 430.8 |
| 578 | 12:03 | 5.32 | 71.6 / 54.6 | 7.7 | 7.1 | 7.7 | 3.7 / 2.3 | 8.04 | 425.3 |

**Table B3** Transect parameter of cruise LP20190515 on RV *Ludwig Prandt* in the Dutch Wadden Sea in May 2019. Shown are values of latitude (Lat), longitude (Lon), temperature (Temp), salinity (Sal), total alkalinity (TA), dissolved inorganic carbon (DIC), silicate (Si), nitrate ($NO_3^-$), the calcite (Ca) and aragonite (Ar) saturation states, and pH per sample. Our salinity and temperature data were complemented by data of three Rijkswaterstaat stations, which were close to our stations. There, Dantziggat* (53°24'4.093", 5°43'37.132") showed temperatures of 11.4 and 14.9 °C and salinities of 31.9 and 31.2 on 10 and 27 May 2019, respectively, Terschelling 10** (53°27'37.318", 5°5'58.129") showed temperatures of 11.4 and 12.9 °C and salinities of 32.8 and 33.4 on 15 and 28 May 2019, respectively, and Vliestroom*** (53°18'48.054", 5°9'35.655") showed a temperature of 11.8 °C and a salinity of 31.1 on 14 May 2019.

| Sample No. | Time [UTC] | Day May | Lat. [°N] | Lon. [°E] | Temp [°C] | Sal [PSU] | TA / DIC [µmol kg$^{-1}$] | Si / $NO_3^-$ [µmol L$^{-1}$] | Ca / Ar [Ω] | pH |
|---|---|---|---|---|---|---|---|---|---|---|
| 535 | 07:56 | 20 | 53.18 | 5.4 | 14.72 | 30.24 | 2507 / 2357 | 10.00 / 5.10 | 3.0 / 1.9 | 7.92 |
| 536 | 08:26 | 20 | 53.19 | 5.34 | 14.81 | 32.51 | 2458 / 2296 | 3.45 / 0.46 | 3.1 / 2.0 | 7.92 |
| 537 | 08:53 | 20 | 53.22 | 5.29 | 14.05 | 32.38 | 2413 / 2227 | 1.92 / 0.85 | 3.4 / 2.2 | 8.00 |
| 538 | 09:28 | 20 | 53.23 | 5.19 | 13.36 | 32.65 | 2381 / 2153 | 0.52 / 1.02 | 4.0 / 2.6 | 8.10 |
| 539 | 09:53 | 20 | 53.27 | 5.17 | 13.17 | 32.65 | 2389 / 2161 | 0.45 / 1.37 | 4.0 / 2.6 | 8.10 |
| 540*** | 10:36 | 20 | 53.33 | 5.12 | 12.77 | 32.97 | 2375 / 2118 | 0.32 / 0.84 | 4.4 / 2.8 | 8.16 |
| 541 | 11:03 | 20 | 53.32 | 5.0 | 12.41 | 33.25 | 2368 / 2097 | 0.34 / 0.77 | 4.6 / 3.0 | 8.19 |
| 542** | 11:49 | 20 | 53.4 | 5.1 | 12.93 | 32.92 | 2374 / 2109 | 0.44 / 0.81 | 4.6 / 2.9 | 8.17 |
| 543 | 12:49 | 20 | 53.45 | 5.33 | 12.95 | 32.45 | 2385 / 2196 | 6.25 / 1.85 | 3.4 / 2.2 | 8.02 |
| 544 | 13:31 | 20 | 53.46 | 5.5 | 13.55 | 32.51 | 2388 / 2169 | 3.02 / 1.30 | 3.9 / 2.5 | 8.08 |
| 558 | 11:33 | 22 | 53.41 | 5.65 | 13.31 | 32.51 | 2384 / 2224 | 2.57 / 1.56 | 3.0 / 1.9 | 7.95 |
| 559 | 12:04 | 22 | 53.4 | 5.66 | 13.45 | 32.51 | 2393 / 2195 | 1.58 / 1.41 | 3.6 / 2.3 | 8.03 |
| 560 | 12:40 | 22 | 53.39 | 5.69 | 13.67 | 32.52 | 2391 / 2183 | 1.52 / 1.33 | 3.7 / 2.4 | 8.05 |
| 561 | 13:09 | 22 | 53.4 | 5.73 | 14.23 | 32.48 | 2418 / 2242 | 2.04 / 1.04 | 3.3 / 2.1 | 7.97 |
| 562 | 13:32 | 22 | 53.42 | 5.77 | 14.71 | 32.51 | 2417 / 2237 | 3.23 / 1.04 | 3.3 / 2.1 | 7.97 |

| | | | | | | | | | |
|---|---|---|---|---|---|---|---|---|---|
| 563* | 13:56 | 22 | 53.42 | 5.82 | 15.33 | 32.45 | 2421 / 2242 | 4.68 / 0.86 | 3.3 / 2.1 | 7.96 |
| 579 | 09:05 | 24 | 53.42 | 5.77 | 15.26 | 32.65 | 2417 / 2215 | 2.71 / 0.86 | 3.7 / 2.4 | 8.01 |
| 580 | 09:31 | 24 | 53.4 | 5.73 | 14.99 | 32.66 | 2426 / 2249 | 3.52 / 1.46 | 3.3 / 2.1 | 7.95 |
| 581 | 10:01 | 24 | 53.4 | 5.66 | 13.87 | 32.58 | 2396 / 2205 | 1.69 / 1.60 | 3.5 / 2.2 | 8.01 |
| 582 | 10:25 | 24 | 53.43 | 5.61 | 14.36 | 32.51 | 2389 / 2193 | 2.23 / 2.52 | 3.6 / 2.3 | 8.02 |
| 583 | 10:59 | 24 | 53.5 | 5.61 | 13.48 | 32.58 | 2382 / 2187 | 3.69 / 3.41 | 3.5 / 2.2 | 8.03 |
| 584 | 11:31 | 24 | 53.51 | 5.76 | 13.50 | 32.59 | 2390 / 2172 | 1.93 / 2.96 | 3.9 / 2.5 | 8.07 |
| 585 | 12:00 | 24 | 53.52 | 5.9 | 13.65 | 32.59 | 2390 / 2173 | 2.34 / 2.23 | 3.9 / 2.5 | 8.07 |
| 586 | 12:30 | 24 | 53.52 | 6.04 | 13.50 | 32.48 | 2384 / 2179 | 2.00 / 1.53 | 3.7 / 2.3 | 8.05 |
| 587 | 13:02 | 24 | 53.45 | 6.07 | 15.13 | 32.40 | 2389 / 2169 | 0.70 / 1.19 | 3.9 / 2.5 | 8.05 |
| 588 | 13:31 | 24 | 53.42 | 6.38 | 15.63 | 31.96 | 2396 / 2182 | 1.33 / 0.62 | 3.9 / 2.5 | 8.04 |
| 589 | 07:20 | 25 | 53.42 | 6.18 | 15.73 | 28.31 | 2430 / 2245 | 4.16 / 5.13 | 3.6 / 2.3 | 8.02 |
| 590 | 07:52 | 25 | 53.44 | 6.09 | 15.80 | 30.90 | 2407 / 2225 | 1.58 / 1.39 | 3.4 / 2.2 | 7.98 |
| 591 | 08:21 | 25 | 53.48 | 6.08 | 15.34 | 31.82 | 2395 / 2222 | 4.16 / 5.09 | 3.3 / 2.1 | 7.96 |
| 592 | 08:51 | 25 | 53.52 | 6.05 | 14.80 | 32.41 | 2386 / 2178 | 0.71 / 0.67 | 3.8 / 2.4 | 8.04 |
| 593 | 09:22 | 25 | 53.55 | 6.18 | 13.96 | 32.30 | 2379 / 2175 | 0.36 / 1.52 | 3.7 / 2.3 | 8.04 |
| 594 | 09:53 | 25 | 53.58 | 6.31 | 13.43 | 32.14 | 2332 / 2148 | 0.34 / 5.75 | 3.3 / 2.1 | 8.01 |
| 595 | 10:24 | 25 | 53.61 | 6.45 | 13.47 | 32.10 | 2347 / 2113 | 0.26 / 5.19 | 4.1 / 2.6 | 8.12 |
| 596 | 11:05 | 25 | 53.58 | 6.63 | 14.50 | 29.99 | 2381 / 2184 | 0.78 / 20.25 | 3.7 / 2.3 | 8.05 |
| 597 | 11:33 | 25 | 53.52 | 6.75 | 14.94 | 29.17 | 2383 / 2214 | 3.04 / 27.84 | 3.3 / 2.1 | 7.99 |
| 598 | 12:00 | 25 | 53.47 | 6.85 | 15.28 | 27.82 | 2395 / 2249 | 8.90 / 37.93 | 3.0 / 1.9 | 7.94 |
| 599 | 12:30 | 25 | 53.4 | 6.95 | 15.46 | 26.39 | 2423 / 2284 | 17.63 / 36.54 | 2.9 / 1.8 | 7.94 |
| 600 | 12:59 | 25 | 53.33 | 7.02 | 15.76 | 23.01 | 2460 / 2343 | 41.93 / 37.68 | 2.7 / 1.7 | 7.92 |
| 601 | 13:29 | 25 | 53.33 | 7.16 | 15.96 | 20.25 | 2517 / 2430 | 56.32 / 37.94 | 2.3 / 1.4 | 7.86 |

444

## Data availability

446 The data of this study are presented in the appendices of this article.

## Author Contributions

MN wrote the manuscript, did the carbon sampling and sample measurement, analyzed and evaluated the data, and led the study. JvB led the research cruise. JvB and HT contributed with editorial and scientific recommendations. MN prepared the manuscript with contribution from all co-authors.

## Competing interests

The contact author has declared that none of the authors has any competing interests.

## Acknowledgement

We thank the crew from RV *Ludwig Prandtl* for their support during the cruise. We thank Leon Schmidt for the nutrient sampling and measurements, Marc Metzke for the C/N measurements, and Yoana Voynova and her department for the FerryBox preparation. We further thank the Editor and two anonymous reviewers for their constructive comments, which greatly improved this manuscript.

## Financial support

This research has been funded by the German Academic Exchange Service (DAAD, project: MOPGA-GRI, grant no. 57429828), which received funds from the German Federal Ministry of Education and Research (BMBF).

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
