# Peer review of "Alkalinity sources in the Dutch Wadden Sea"

_EGUsphere, 2023_

## Author Comment (AC1)

**RC1**: 'Comment on egusphere-2023-2595', Anonymous Referee #1, 14 Dec 2023 reply

**General comment**

The manuscript "Distribution and source attribution of alkalinity in the Dutch Wadden Sea" examines spatial and temporal alkalinity dynamics in the Dutch Wadden Sea. The topic is timely and fits the scope of Ocean Science. The study design is appropriate, and the data set sufficient to answer the stated research questions. However, the manuscript is hard to follow and should be streamlined. The entire manuscript should have 3 to 5 well-structured paragraphs per page that address one specific topic. The introduction reads more like a study site description and does not give a broader context or demonstrates the need for this study. It would be helpful to put some of the results (e.g., most subplots of Figure 2 and Table 1) into a supplementary information and decrease the overall word count by focusing only on most relevant findings. The authors should pick up to three key messages on which they focus through the manuscript. The figures need some improvements. I suggest accepting the article after major revision.

*AC: Dear reviewer, thank you very much for your positive and helpful comments, which have really improved our manuscript. In line with your suggestions, we have revised the Introduction, restructured the Methods and Results sections and also revised and improved the figures. Our detailed responses can be found below. As the study of TA generation and its possible pathways is a complex topic, greatly reducing the word count and removing parts of the manuscript would result in an incomplete study approach. We think that using at least all the parameters/stoichiometries we have completes the approach to identify TA-generating pathways. Even if this leads to a more complex and complicated discussion, we would like to stick to this more complex approach. However, we have revised some of the sentences in the discussion to make them more understandable.*

**Specific comments**

Consider a more interesting and specific title.

*AC: Done. We have adjusted the title in a shorter and more specific version.*

L12 remove: "and compared it with earlier data".

*AC: We would like to stick to the announcement of the data comparison in the Abstract. However, we have rearranged the sentence in line with RC2 and changed the term "earlier" to "historical" data.*

L19-22. This sentence is hard to follow and has too many citations. Consider splitting your sentences into two if they span over several lines and only cite most relevant literature for each statement.

*AC: We have revised the Introduction and some sentences, such as the one mentioned above.*

L64 – 66 Add dates.

*AC: Done.*

L75 Which kind of carbon measurements?

*AC: We have replaced carbon with the specific carbon terms TA and DIC.*

L137 Remove first sentence of caption.

*AC: Done.*

L201 Remove sentence. Avoid writing "various parameters "through the manuscript and list parameters instead. Try to be more specific.

*AC: We have deleted the sentence you suggested and clarified the parameters where it makes sense to do so.*

L204-205 Explain how you get to this conclusion.

*AC: This sentence is based on plots not shown. We have edited the sentence.*

L309 Faber studied mangroves. There is probably a study that is more similar to your study site.

*AC: It is true that the study site in the reference is different from ours. However, what is interesting here is not the location, but the use of a tracer, in this case radon, to identify the source of higher TA values in pore water discharge, for which we used silicate instead of radon.*

It is very hard to follow your discussion. Suggest reducing to the most interesting findings. Condense the information as much as possible.

*AC: As mentioned in our first statement above, we would like to stick to all parts of our discussion. However, we agree with the complexity and have revised some sentences of the Discussion to make it more understandable.*

Figures and tables:

Fig 1: Remove ESRI source code (in all figures).

*AC: In accordance with the Ocean Science's submission guidelines, we would like to stick to the ESRI source code in Fig. 1. However, Fig. 2 has been fully edited and created in R instead.*

Fig. 2: This figure is too big. Show only one figure per page. Choose only most important parameters and put rest into the SI. You could plot all parameters against salinity and/or distance to the coast in one graph with subplots and discuss trends.

*AC: We would like to stay with the spatial representation, but we have revised Fig. 2 and included most of the parameters in the Appendix according to your suggestion.*

Fig. 4 and 5: Increase font size and point size. Axes should be aligned. Salinity looks strange in Fig 4d.

*AC: Fig. 4 has been revised for better readability.*

Put Table 1 in SI. Could add averages and standard deviations.

*AC: Table 1 has been moved to the Appendix. As Table 1 contains individual observations, average values and standard deviations are irrelevant.*

**Technical corrections**

When you reference two or more citations, the single citations are separated by a semicolon without any spaces. This looks strange. Please double check the citation guideline of the journal.

*AC: The citation was made using the reference manager EndNote and the Copernicus style file "Copernicus Publications" as output style, as indicated on the Ocean Science submission website.*

Avoid paragraphs with only one or two sentences.

*AC: We have deleted some very short paragraphs and also restructured the section on methods section a little.*

L81 Remove "." before citation.

*AC: Done.*

L191 TA was defined before.

*AC: We are not sure what you mean by this comment, as L191 contained Table 1 in the previous version of the manuscript.*

L198 Rephrase this sentence, "arrived" does not make sense in this context.

*AC: Thank you. We have changed "arrived" to "obtained".*

Do not overuse the term "shed light on…"

*AC: Some of the terms have been revised to reduce the use of "shed light on".*

The reference list needs to be edited. E.g. $CO_2$ with subscript and uniform styles for titles. Include volume and page numbers for all references.

*AC: We have revised the reference list.*

---

## Author Comment (AC2)

**RC2**: 'Comment on egusphere-2023-2595', Anonymous Referee #2, 20 Jan 2024 reply

Strengths:

I really appreciate that the authors left the location names on Figure 2 for easy reference when going from reading the text in the results to the figures. The manuscript as it stands is well organized throughout and easy to follow. The discussion of the results and considerations is also quite comprehensive and strong, aside from the one reservation I have that is listed below. I suggest accepting it after major revisions.

*AC: Dear reviewer, Thank you very much for your very detailed comments, which have really improved the manuscript. You will find our responses below.*

Major Concerns:

Results:

Figure 2: Please make points in figure 2 a color gradient from min to max value instead of grouping many values in one color. That way the reader can more specifically see the actual value at each site. The groupings also aren't consistently spaced which could be misleading when analyzing spatial trends. Also, this is entirely a personal preference but generally I think blue=lower and yellow=higher based on the ocean of things color bars so it was confusing at first and they may be better reversed.

*AC: Thank you for highlighting this. We agree with your suggestions and have revised Fig. 2 by reversing the color scale and adding a color gradient. As suggested by RC1, we have split Fig.2 and shown only the most important plots in the Results section and moved the other plots to the Appendix.*

TA generation section: I think the assumption that TA generation for the entirety of the Wadden Sea does not come from freshwater dilution/river water is not properly supported as stands and is to general. I agree though that this is the correct conclusion for the Ameland region based on the data you've shown in Table 1, Figure 4, & 5. However, the authors previously show in Figure 3 that the TA vs salinity trend for the Ameland region is non-conservative, while the Ems-Dollard & Vlie regions of the Wadden Sea are conservative. Therefore, TA & salinity trends are clearly not consistent across the entire region. Which may be attributed to varying degrees of freshwater discharge across the region, which they mention briefly in the discussion. So while it may be true for the Ameland region that TA sources do not come from freshwater dilution/river water, the same cannot be said for these other regions. If the authors had stationary data during ebb tide for the regions that do appear to have a conservative relationship and saw the same pattern then yes they could reasonably assume this is true for the entirety of the Wadden Sea on average. However, they do not appear to have stationary data for either of the conservative regions. Therefore, this statement needs to be more specific to the Ameland region of the study and not the entire Wadden Sea region. If there are river gauges in the region to support similar river discharge across the region then this assumption may have stronger support but would still be rather bold without ebb tide data from the other regions. They also say later that freshwater dominates in the EMS-Dollard.

*AC: Thank you for your detailed description. We agree that an overall statement for the whole Wadden Sea, including areas with freshwater input, could lead to misleading*

*interpretations. For this reason, we also pointed out in the last part of the Discussion that a comparison between the northern Wadden Sea and the Dutch Wadden Sea is not advisable. To make this even clearer, we have clarified our statement.*

Minor Concerns:

Throughout manuscript: Please remove any we's, he's, etc. Instead refer as "this study" "the data" etc.

*AC: As this is a matter of taste, but the use of pronouns in the first-person is accepted, we would generally like to stick with it. However, we also feel that it has been used too often and significantly reduced its use.*

Methods:

Is there a reason you focus on calcite saturation instead of aragonite? Is it more relevant to local species? Please state.

*AC: There was no particular reason why we only reported calcite. Therefore, we have added the aragonite values to complete the data.*

Please include how you generated the statistics throughout the paper (i.e. the linear regressions). Did you use Excel, Matlab, R, etc? Also, are these Model I or Model II regressions?

*AC: Thank you for pointing this out. We have added a section on Data analyses and also corrected the linear regressions from Fig. 5 to a Model II.*

Discussion:

Lines 237-239: Could this also be attributed to seasonal differences or changes in watershed usage over those 30 years?

*AC: We can rule out a strong seasonal influence here, as both studies (Hoppema's and this study) took place in May. However, Hoppema's study took place at a time of high eutrophication and high local primary production (Cadée & Hegeman, 2002), and we cannot rule out an increased influence on the catchment and freshwater inflows. However, we have not focused so much on the freshwater influence here, but on TA generation in the sediments of the tidal flats. We have revised this section and will also include a statement to make this point clearer.*

Minor Edits:

Line 10: "North Sea is hypothesized to be a source…"

*AC: Done.*

Line 10-11: "This study measured TA…"

*AC: The sentence was reworded to "This study observed..".*

Line 12: "compared is with historical data."

*AC: Done.*

Line 14: " washed out with outgoing tide water."

*AC: Done.*

Line 27: "Most of the Wadden Sea is located..."

*AC: Done.*

Line 28: "which makes it the world's largest uninterrupted stretch…"

*AC: Done.*

Line 31: what dynamics? Biogeochemical? Chemistry?

*AC: We have added the term "biogeochemical".*

Line 30-33: Split into 2 sentences. One for chemical and one for physical sources of variability.

*AC: Done.*

Line 35-36: what water masses? What is a 'strong' exchange? Do you mean that the water masses are very different?

AC: We have deleted the "strong" and revised the sentence to make it clearer.

Line 37: "The carbon storage capacity of the North Sea is an important atmospheric CO2 sink as it exports and stores the absorbed…"

*AC: Done.*

Line 40-43: TA, primarily consisting of bicarbonate and carbonate, is generated by chemical rock weathering (citations), calcium carbonate dissolution, and anaerobic metabolic process, such as…"

*AC: We have revised this sentences according to your suggestion.*

Line 54-56: during what time of year?

*AC: We have added the time (May).*

Lines 116-118: please include aragonite saturation values as well. They can simply be added next to calcite values in parentheses.

*AC: Done.*

Lines 204-205: Please include the R-values for the relationships to back this up.

*AC: Done.*

Lines 215: "by enhanced water movement" what does this mean? Be more specific.

*AC: There, the water movement is driven by tidal forcing. We have added this explanation to the text and also removed the word "enhanced".*

Line 219-220: "which could be traced back on an effect of the first four samplings as mentioned above" move to discussion

*AC: This is only the second part of the sentence. The first part is related to the results, which is why we want to leave it in its original place.*

Line 221: connect this back to higher denitrification and ammonium production compared to nitrification because your ammonium relationship supports this as well.

*AC: We have discussed this in the Discussion section 4.3.2.*

Line 243: A TA increase of >70 is quite a bit. I wouldn't say it's only slightly higher.

*AC: Yes this is true. We meant it more as a comparison. However, we removed this sentence by editing this part anyway.*

Line 271: Where is Marsdiep relative to your study sites? Please add to map.

*AC: Since Marsdiep is much further west, it is not visible on the maps, and adding it would enlarge the map too much.*

Lines 282-287: I think these sentences probably belong in the discussion.

*AC: This is in the Discussion.*

Lines 285-287: and more spatial data for the varying TA vs salinity relationships of the different tidal regions in addition to the temporal data mentioned

*AC: We have revised the conclusion and added a corresponding explanation.*

Line 291: this is not necessarily true. Increased agriculture has led to increased rock and soil exposure, resulting in increased rates of silicate weathering. Also, I'm not sure if this applies to this region but digging of quarries also extracts and exposes silicate minerals during the mining process. However, if the region has little influence from either of these then it could be true. Apologies for I don't know what the land use is like for this region.

*AC: In general, it can be assumed that Central Europe and the river catchment areas mainly contain carbonate minerals. High turbidity concentrations in rivers and also the TA concentrations in the rivers, which are far above the general level in the North Sea, indicate a dominance of carbonate minerals and speak in favor of this. The TA generation by silicate weathering is also a very slow process.*

Line 290: "insight" not "inside"?

*AC: Done. Thank you!*

Line 317: what about the ems-dollard region?

*AC: The conservative behavior in the outer Ems-Dollard Inlet is mentioned above in the Results (Fig. 3) and in the Discussion section 4.1, where the spatial distribution of TA is discussed. It is true that TA is generated in the upper / tidal river of the Ems Estuary (Norbisrath et al., 2023). However, this site is located further east and shows clearly decreasing TA values with increasing salinities. The focus here is on TA generation in the tidal flats between the barrier islands and the mainland, and in particular on our study site around Ameland island, where we observed constant marine salinities (<30).*

Line 403: what necessary data? This is a good place to suggest future work.

*AC: Since the Wadden Sea is a well explored area, we cannot clearly say what has been done and what has not, especially with regard to sediment-related work. However, for the surface and water column TA, we have suggested some future work (e.g. seasonal and end-member observations) in the Conclusion.*

---

## Author Response (AR2)

**June 06 2024 Editor comments:**
**Public justification (visible to the public if the article is accepted and published)**:

*Dear Editor,*

*Thank you very much for your time, work, and the helpful comments and suggestions, which really improved our manuscript. We significantly edited the manuscript in accordance to your suggestions. We reworked the Introduction and reduced the Discussion in length. We also rephrased many sentences to increase the readability.*

Dear authors,

There is obviously a big difference between the two subregions around Ameland and the Ems estuary. This should be made more clearly and separated in the manuscript.
*AC: We have addressed this concern and have reworked several parts of the manuscript to make this point more clear.*

The Introduction still needs work and restructuring, even if some changes were already made after previous rounds of reviews.
*AC: We have reworked the Introduction.*

It is mentioned that two (half) tidal cycles were sampled and measured. However, data of only one of these are shown and discussed. Do the other data show the same features or are they quite different? This is important info which must be provided. Please mention somewhere what the other data look like, how they compare with the presented data and whether the conclusions are supported by the second half tidal cycle.
*AC: We added a statement in the Methods section saying that we only used the second half tidal cycle (flood tide) for a data comparison and to relate whether the ebb tide data are in a similar range. We added the flood tide data in the Appendix.*

A previous review objected against the use of contents instead of concentration. However, the use "contents" instead of concentration is allowed. Maybe cite the paper by Jiang et al (2022, Front Mar Sci), to make this clear.
*AC: We have changed it back to the use of concentration that we already used in the first version.*

The notation 2400 umol kg-1 TA, i.e., with TA at the end is not common and may be confusing. Please change it throughout the manuscript for all concentrations/contents.
*AC: We had this notation in the first version of the manuscript and changed it back to µmol TA kg$^{-1}$ etc.*

List of minor and technical issues:
L10 "The oceanic buffering capacity total alkalinity (TA), as the major global CO2 sink, is of growing scientific interest". This is a strange contention and not necessarily true. Please change, for example, by just saying that TA is an important chemical property which plays a role in oceanic buffering capacity.
*AC: We agree, and changed the sentence according to your suggestion.*

L11 "… generated by chemical weathering on land …" Add: on land.
*AC: Done.*
L12-13 This study shows observations of TA, … instead of "This study observed TA, …"
*AC: Done.*
L18-19 "We assume that seasonality and the associated nitrate availability in particular influence TA

generation by denitrification, which we assume is low in spring and summer." This is too much of assumptions in an abstract. Please only present the facts.
*AC: We have rephrased this part.*

L24 "The (still unofficial) Anthropocene describes …" Or something similar, as the Anthropocene has not been acknowledged officially.
*AC: We added a 'so called' and "Anthropocene".*
L25-26 "The climate and the increasing atmospheric CO2 content is mainly regulated by the open ocean." This is not correct. The biosphere takes up more anthropogenic CO2 than the oceans – see the Global Carbon Budget.
*AC: We removed the 'mainly'.*

L26 The around 30% is more like 25% (see Global Carbon Budget).
*AC: We have corrected it.*
L32 susceptible to what? Please be more precise.
*AC: We added "to changes".*
L49 strong tidal currents instead of: high tidal currents
*AC: Done.*
L55-57 "Understanding of TA sources have recently become increasingly important due to increasing anthropogenic CO2 emissions, and the resulting demand for ocean based net-negative CO2 emissions" This needs a short explanation as the connection between the two is unclear.
*AC: We rearranged this sentence into two and added a connection to make it more comprehensible.*
L74 Please use format like 21 May 2019
*AC: Done.*
L76 delete: continuously
*AC: Done.*
L80 Add solution after chloride
*AC: Done.*

Section 2: When were the samples measured, i.e., how long after sampling? Please add this to the different methods.
*AC: We have added the reference material and the month when the samples were measured.*
L97 I guess you mean: … both with a measurement precision …
*AC: Yes, thank you. We adjusted the sentence.*
L97 It is kind of strange that you cite the precision with a different study. Did not you yourself determine the precision and accuracy?
*AC: This mentioned study references the methods and the official given precision of the instrument.*

L98-99 "To ensure a consistent calibration of both measurements, certified reference material (CRM batch # 187) provided by Andrew G. Dickson (Scripps Institution of Oceanography) was used" How was this CRM used? Did you adjust the measured data?
*AC: We added the missing information. CRM were measured before and after the samples and used for data drift correction.*

Section 2.3 What is the precision and accuracy of the nutrient measurements? Were any CRMs used?
*AC: We have added the used reference materials and the max. standard deviation.*
L117 change to: (in particular Hoppema, 1990)
*AC: Done.*

L117-118 " … we observed TA and related parameters from the coastal mainland towards the open North Sea as a surface water transect" (delete: "the spatial distribution of" because that was already mentioned earlier in the sentence.

*AC: Done.*

L119-120 „Salinity was relatively stable with only minor differences varying from 28 to 33" First, I think stable is the wrong term here. Second, this large range does not indicate a homogeneous salinity but rather significant variations. The range of salinity is obviously larger than this, as values in the Ems are smaller. Please rephrase the whole paragraph.
*AC: Done.*

L123 "oceanic" is not the correct term here. The North Sea is not an ocean. Actually, "In contrast to the oceanic side," may as well be deleted as it has no function here.
*AC: We changed all the "oceanic" into North Sea side.*
L124 "Only in the Ems Estuary, the contents were even higher" Delete: Only. Also close to Harlingen TA is higher.
*AC: Done.*
L126 support
L126 "supporting the assumption of TA being generated in this tidal flat area." It is too early for this contention. It may as well be the discharge of river water that causes high TA.
*AC: We removed this sentence.*

L127 "showed a similar pattern" Similar to what? Please phrase more precise.
*AC: We removed this part.*
L127 Use other word for "ocean" Please also change this at other places in the text.
*AC: Yes, we did this and used just the term "North Sea".*
L129 I think the silicate contents are given with one digit too much, as the accuracy is probably not that high (accuracy has to be stated in the Methods section)
*AC: Done.*
L130 not site, rather region
*AC: Done.*
L134 delete: similarly
*AC: Done.*
L135 region instead of transect. There are a few transects shown in this study.
*AC: Done.*
L144 "The strong impact from the inner Ems Estuary is visible in all parameters with higher values in the outer estuary and its adjacent zones" This is an awkward sentence. Please rephrase.
*AC: Yes, we rephrased this paragraph.*
L145 higher values of what?
*AC: We rearranged this whole paragraph.*
L148 in the mixing plot between TA and salinity. Add: plot, namely TA and salinity do not mix themselves.
*AC: Done.*
L148 "A relatively linear mixing behavior" Relatively is awkward here. Please rephrase
*AC: Done.*
L150 "identifying the Dutch Wadden Sea as a source of TA" This is not correct. A linear relation in an estuary in this case shows that the TA of the river water is high.
*AC: This is correct. This part of the sentence was only related to the Vlie Inlet. However, we rearranged the paragraph since it led to misunderstanding.*

L150-151 "In contrast to the TA content computed for the salinity end-member in the Ems-Dollard Inlet, we detected higher TA contents around Ameland" This sentence is strange; please be more precise.
*AC: Both this sentence and the whole paragraph were rearranged.*
L154 The values are not increasing. There is just a range of values at constant salinity.

*AC: Done.*
L156 (figure caption) Change to: Mixing plot of total alkalinity (TA) and salinity …
*AC: Done.*

Figure 3 Please indicate whether the correlations shown are statistically significant or not.
*AC: Done.*
L169 "DIC contents were similar to TA" This is of course not correct, as the data show. This is about the development/course of DIC contents.
*AC: Yes, we adjusted this sentence to make it clearer.*
L172 "Nitrate concentrations approached seawater concentrations" What are seawater concentrations for nitrate? These are highly variable. Please be more precise in what you intend to say.
*AC: We have reworked this part.*

L173 delete: slightly
*AC: Done.*
L183 Change to: the maximum pH was 8.07 at high tide
*AC: Done.*
L188 (caption) This is only half a tidal cycle. Maybe it is good to mention the local time difference with UTC
*AC: Done.*
Figure 4: Two important variables are missing, namely salinity and temperature. These must be added here for better understanding of the tidal cycle.
*AC: Done.*
L193 "The 192 strong tidal forcing induces a strong benthic-pelagic coupling" This needs a reference.
*AC: Done.*
L194 hypothesis instead of assumption appears to be the better choice here.
*AC: Done.*

L201-202 "Based on the correlation of TA and silicate (R2 = 0.93), and on the nonlinear relation between both, TA and salinity (R2 = 0.32), as well as silicate and salinity (R2 = 0.21)," How can a non-linear relation have a correlation? And is this correlation significant or not? This must be rephrased to something like: Based on the correlation of TA and silicate (R2 = 0.93), and on the insignificant relation between TA and salinity (R2 = 0.32), as well between as silicate and salinity … .
*AC: This is why we named it non linear relation (and not correlation), but we agree that this can be misleading and adjusted the sentence accordingly to your suggestion.*

L203 "Both TA and silicate increased almost proportionally" What does this mean? Please describe more clearly.
*AC: We have rearranged this sentence.*
L204 Change to: non-conservative behavior of TA with respect to salinity and silicate to salinity …
*AC: Done.*
L207 "from high tide to low tide" I guess you just mean the half tidal cycle as treated above. As now, this is confusing.
*AC: That is correct, so we added "the half tidal cycle".*

L208 "recommend" here is awkward. And why would you ignore these data? This sounds quite arbitrary. Please give a good reason for that.
*AC: These data are not ignored in the analyses. We just wanted to highlight that the first points are on the tipping point between the change in tide. We deleted the confusing part of the sentence and reworked this section.*

L209-210 "First, the correlation between TA and DIC reveals the ratio between anaerobic and aerobic

processes, which identifies a strong positive correlation between DIC and TA (R2 = 0.93) with TA contents higher than DIC contents (Fig. 5a)." This is not well phrased. Change to something like: The correlation between TA and DIC is a measure for the ratio between anaerobic and aerobic processes. Our data show a strong positive correlation between DIC and TA (R2 = 0.93) with TA contents higher than DIC contents (Fig. 5a).

*AC: Thank you for this suggestion. We arranged the sentence accordingly.*

L210-212 "However, even with contents of TA higher than DIC, the slope of 1.87 indicated DIC release excess with an increase in DIC (ΔDIC = 101.3 µmol kg-1) almost twice as high as TA (ΔTA = 51.6 µmol kg-1) (Fig. 5a)." This sentence is unclear. Please rephrase so that it becomes clear what is meant here.

*AC: Done.*

L214 delete: almost

*AC: Done.*

L218-219 "suggest a stronger effect of TA generation than nitrate production". The word "effect" can be omitted here as it is confusing: … suggest a stronger TA generation than nitrate production …

*AC: Done.*

L239 insert possible or probable before TA

*AC: Done.*

L243 add something like: …similar range, but the spatial gradients are opposite.

*AC: Done.*

L244-246 "The conservative mixing in the Vlie Inlet can be explained by the fact that more North Sea water pass through the deeper inlets and transport more seawater towards the coast." This is not clear to me. How does this explanation work?

*AC: We have rearranged this sentence.*

L257 The salinities observed were not constant. They varied possibly not that much, but that is not the same as constant. Please rephrase.

*AC: Done.*

L264 hypothesized instead of: assumed? Did they find indications for it?

*AC: We have changed it into "indicated".*

L266 "from the early 1990s" The study was actually from the late 1980s.

*AC: Done.*

L268-269 "Our present TA data and the historical TA data show no large differences in the range of values observed during a tidal cycle." This sentence is strange. What about: Comparing our present TA data and the historical TA data, there is not a large difference in the range of values observed during a tidal cycle.

*AC: Done.*

L269-270 "However, an in-depth interpretation and comparison of both data sets would exceed the capacity of these data, leading us to focus on TA generation during our cruise" I do not understand this sentence. What is the capacity of these data?

*AC: Here, we mean the limited number of TA data. We have rephrased this sentence.*

L275 estimated instead of assumed. It would be good to add that they used a model for this at this place.

*AC: Done.*

L276-277 "However, an inclusion of our TA export into the model used by Schwichtenberg et al. (2020) would be unreliable, since our TA export based only on one tidal observation" Why should that be unreliable? It can be just part of that estimate, as the present data are observed data. However, if you would use the present tidal data for estimating the entire TA export, that indeed would be unreliable.

*AC: Yes, that is what we wanted to say. We have rearranged these sentences.*

L283 "In order to gain further insight into potential sources of TA, we included nutrients in our investigation" This sentence is too general. Earlier in the manuscript, nutrients were already shown.
*AC: We have rephrased this sentence.*

L285 Write: Van der Zee, or van der Zee
*AC: Done.*

L283-285 "The main focus was on silicate that we used as a natural tracer since it is not directly provided anthropogenically and allowed us to determine the silicate source" This sentence is unclear; please be more precise what you want to convey. The authors base this contention on a reference from 2005. Since the early 2000s the situation in the Wadden Sea could have changed a lot, as was also found for the sources of alkalinity. Do you have any indication that nothing has changed?
*AC: We have rewritten this sentence to clarify why we used dissolved Si as a tracer for pore water exchange and substantiate this with early work on Si dynamics in the Dutch Wadden Sea. Details on pore water data were later published by Rutgers van der Loeff (1980) but already used by van Bennekom et al. (1974). The question whether Si dynamics have changed is valid. River data do not show a large change in dynamics, but in our opinion, an in-depth discussion of the Si long-term dynamics is beyond the scope of this article.*

L288 hypothesis instead of assumption
*AC: Done.*

L290 "salinity consistently being above 32" It is not clear which area is concerned here. The Ems estuary has lower salinities, but also the Vlie has a salinity of about 30 (Fig.3). Please be more precise.
*AC: We have added the sampling area (around Ameland island).*

L291-292 "This can be supported by the relation between silicate and salinity in which we observed a 291 non-conservative behavior (Table B1). Since TA behaves also non-conservative relative to salinity (Table B1)" This says actually the same as the previous sentence. In addition, Table B1 does not show a relation at all; it just shows data. You may just refer to Table B1 after the previous sentence.
*AC: We have rearranged both sentences.*

L297-298 delete: "In May, they observed low salinities indicating freshwater. However, in September they observed constant marine salinities referring them to be exclusively composed of recirculating seawater."
*AC: Done.*

L298-299 "Since we constantly observed marine salinities," This is an awkward sentence. Change to something like: Since we observed relatively constant salinities …
*AC: Done.*

L307 "smaller than" instead of: below
*AC: Done.*

L308 delete (twice) clearly
*AC: Done.*

L311-313 One can write this much easier and understandable. Please rephrase.
*AC: Done.*

L313-314 "However, when silicate occurred dissolved in water it does not contribute to TA generation (Meister et al., 2022)." I do not understand this. Please explain .
*AC: This sentence was deleted while reworking this part.*

L314-315 "A TA excess related to silicate was also observed in the correlation between TA and silicate (Fig. 5b)." Of course, you can see this in Fig. 5b. These are the same data.
*AC: We have deleted this sentence and added the ref of Fig 5b to the first sentence.*

L318 Use the Greek symbol for omega. Actually, "With the observed omega values," can be deleted because the same is written later in the sentence.

*AC: Done.*

L321 "because of the Ω supersaturation of the overlying water" This is not the reason for not being able to determine TA generation in the sediments, so please delete.

*AC: Done.*

L326-327 This is an anacoluthon. Please correct the sentence. Which ΔTA and ΔDIC is this? They have not been defined.

*AC: Done, we have rephrased this paragraph.*

L328 increase of DIC. Which increase? Again, ΔDIC was not defined. Please be more precise.

*AC: We added "during ebb tide" to show that this is the increase of DIC during ebb tide.*

L330 "high CO2 export" Which export, from where to where?

*AC: Since the word export was misleading, we have changed it into production.*

L330 delete: also, as this is not observed in this study. It is only hypothesized.

*AC: Done.*

L335 continues instead of: continuous

*AC: Done.*

L3345 exceeds instead of: exceeded

*AC: Done.*

L353 less instead of: lower

*AC: Done.*

L357 "due to high turnover rates" I would say: due to high photosynthetic activity

*AC: Done.*

L373 It would be a nice service to the reader and make the paper more readable if the chemical reaction scheme of aerobic OM respiration with the associated formation of ammonium would be shown here.

*AC: In order to reduce the discussion in length as suggested, we removed this whole part of the discussion and reworked it.*

L375 delete: leading to ammonium formation. This was already written at the beginning of the sentence.

*AC: Done.*

L376 increases instead of: would increase

*AC: Done.*

L388-389 "the simultaneous increase of TA and nitrate is noticeable for us, because nitrification consumes TA" It is unclear what the authors try to convey here. Please rephrase and explain.

*AC: We have rearranged this sentence.*

L391 "Low nitrate concentrations resulting in a reduced availability of bound oxygen, i.e., electron acceptors." This is not a correct sentence. Please rephrase.

*AC: We removed this sentence and reworked this whole paragraph to reduce it in length, as suggested.*

L401 define POC, as it is used for the first time

*AC: Done.*

L413-414 "While observing the spatial TA distribution and TA generation" This manuscript is does not observe TA generation per se. It only deduces it and speculates about it.

*AC: We have added "deducing" to this sentence.*

L415 delete: clearly

*AC: Done.*

L420 deduced instead of: identified

*AC: Done.*

L420 and further: It should be made clear that this was not observed but rather deduced with some speculation.
*AC: Done.*

L428 Spatial instead of: Latitudinal and longitudinal
*AC: Done.*

L431 Table B1 It is half a tidal cycle. Please mention the position in the table header. An obviously missing variable in this table is: time.
*AC: We added both.*

L432 Format 21 May 2019
*AC: Done.*

L437 Which other data were measured during the cruise? At least it was mentioned that there is a second (half)tidal cycle. The data should be published along with the manuscript. Other data, which is probably not that much, could be published as supplementary material.
*AC: During this cruise we only measured the analyzed data and the other half tidal cycle, which we used to compare and relate the range of data from the ebb tide tidal cycle. We added these data in the Appendix as Table B2.*

Additional private note (visible to authors and reviewers only):
This manuscript has had a few rounds of reviews already, but it still needs work to get it into shape for final publication. I have listed below quite a long list of mostly technical issues, and the list is not even complete. The writing is partly sloppy and should be much improved. I encourage the senior authors to thoroughly go through the manuscript again. At many places the phrasing is not precise enough for a scientific study. Sometimes the reasoning is hard to follow. In many places the text is overly drawn out and it can be more succinct.

As to the science, I think the data and its discussion are worthy of publication. However, the discussion on the alkalinity sources is in many aspects speculative and as being speculative, it should be reduced in length. Note that there is not a single measurement that could confirm the outcome of the analysis. Thus, the authors should tone down the firm conclusions, even if there are indirect indications that the tidal flats may be important sources of TA. I would still want to give the authors the chance to improve the manuscript and make the major revisions that I am here suggesting. Please note that all comments must be addressed satisfactorily and if the authors decide not to apply the changes, they need a good reason for that.

---

## Author Response (AR3)

Minor revisions by Editor:

**Public justification (visible to the public if the article is accepted and published)**:
Dear authors,

The manuscript has much improved since the previous submission. However, there are still some issues that must be accounted for. Most of these are of technical nature, but not all. My comments are listed below:

*Dear Editor,*
*Thank you very much for your comments. We have adjusted the manuscript according to your suggestions. You will find our point by point responses below.*

L81 Temperature and salinity what precision/accuracy? Was the ferry box calibrated?
I guess the salinity values are on the practical salinity scale. If yes, please mention this.
*AC: Yes, salinity is on PSU. We have added this and the accuracy to the manuscript. The accuracy for the FerryBox is 0.1 °C for temperature and 0.02 PSU for salinity (Petersen et al., 2011). The Ferrybox was cleaned and the system was checked prior to the cruise. Salinity is occasionally checked using discrete samples, which is considered sufficient for gradients in near-shore investigations (personal comm. Y. Voynova). The salinity and temperature data were noted at time of sampling and used for measuring TA and DIC. In addition, we complemented our data with salinity and temperature data observed during Rijkswaterstaat monitoring during May at the stations Vliestroom, Dantziggat and Terschelling 10, showing values in the same range.*

L110 „VKI SW4.1B (NOx, NO2 and NH4) and VKI SW4.2B (Si and PO4)" Any more info on these standards? Manufacturer?
*AC: These are Eurofins standards. We have added the manufacturer info in the manuscript.*
L111 What is a maximum standard deviation?
*AC: The maximum standard deviation is the highest standard deviation of all samples measured for each parameter.*

L128 "with salinities showing only minor differences varying from 28 to 33" I think differences is the wrong word here, it should be variation. And values between 28 and 33 is not minor.
*AC: We have changed it into "smaller variation".*
L185 Use symbol for omega
*AC: Done.*

Fig. 4b The scale for salinity is too large. It prohibits to see any variation, as can be seen in all other sub-panels of this figure. The labels with 3 digits are not correct considering the salinity data given in the tables.
*AC: We have changed the panel scale to 2 digits.*

Again Fig. 4: The distribution of data points of pH, calcite and aragonite looks strange. It is necessary to provide the uncertainty of these variables, preferentially in the methods section.
*AC: Thank you for highlighting this. We have checked the data again and detected an output error in R. We have corrected the pH plot, which matches the omega plot now. We added reported uncertainties for the calculated parameters by Millero et al. (1993) and Orr et al. (2018).*

L310 TA generation is not observed in a strict sense, but rather deduced.
*AC: We have changed "observed" into "deduced".*
L310 "The observed TA generation of 7.6 µmol TA kg-1 h-1 and the silicate increase of 1.4 µmol Si L-1 h-1 indicated an excess of TA" It is not clear what this sentence wants to convey. What is the excess, excess of what?
*AC: This means an observed excess of TA compared to Si.* We have added this to make it clearer.

L311 "A given TA:Si ratio of 2:1" This needs some explanation, i.e., what ratio, why can you use it? Is the mentioned ratio well constrained? Generally this is not the case; there is quite some variability in such ratios.

*AC: We have clarified and softened this sentence.*

L317 "in the water column", instead of: "in the overlying water"
*AC: Done.*
L317 delete: clearly
*AC: Done.*
L325 delete: both
*AC: Done.*
L326 The figure says 1.89, not 1.87
*AC: Done.*

L325-327 "The correlation of DIC and TA reveals an excess of released DIC compared to TA (Fig. 5a), as indicated by the slope of 1.87, while we observed an increase in DIC (ΔDIC) almost twice as high as in TA (ΔTA)." This sentence can be much shorter. The first part essentially says the same as the second part. I think here the use of excess is confusing.

*AC: We have rephrased this sentence to reduce it in length.*

L327-326 "The high ΔDIC points to high aerobic OM degradation and remineralization, resulting in high CO2 production." I think this is not correct here. A slope of about 2 is just the normal case when CaCO3 is involved.

*AC: A slope of 2 only indicates CaCO3 involvement if the ratio used for the calculation is TA:DIC. In our case, this would lead to a ratio of 0.5 as we observed almost twice as much DIC than TA. Here, we used the ratio of DIC:TA. So we defined the ratio in the opposite way. As a support for the CO2 production, we have also added the pCO2 values.*

L339 It is more correct to write a ΔDIC:ΔTA ratio.
*AC: Done.*
L341 delete: potential
*AC: Done. And also in the following lines.*
L341 … of the observed ΔDIC during the tidal cycle … (add explanation that ΔDIC is from this source)
*AC: Done.*

L342 " … the expected Redfield ratio of C:N (6.6) …" I do not assume that this ratio is expected. This ratio is quite variable. Please provide a reference that this ratio is valid in this region.

*AC: For this estimation we have chosen the theoretical expected ratio of marine C:N. Hickel (1980) studied the seasonality of seston composition in the northern Wadden Sea and observed a clear seasonality suggest a strong impact on the SPM composition during summer including minimum C:N ratios during summer of 7.5 (Mol/Mol) and 14 in winter. Given that both refractory SPM and fresh OM contribute to the overall composition of SPM during summer is safe to assume that fresh OM part of SPM has a C:N ratio close to Redfield.*

L361 "in the water column", instead of: "in the overlying water"
*AC: Done.*
L365 "We support their findings of lowered TA generation by denitrification in late spring and early summer." You did not prove lowered TA generation, you actually only deduced it based on results from other scientists in their papers. The sentence cannot be included as is.

*AC: We deleted this sentence.*

L387 "the northern and the western parts of the Wadden Sea" Please be more specific, which parts do you mean?

*AC: We added a more specific description.*

L414 Table B1 What exactly are rounded up values? How were they rounded up and why? Why not just give the data as they were measured. For oceanographic purposes it is strange to give salinity by only one digit. Even a ferry box must be able to measure it with greater precision. For the other parameters the number of digits seems to be ok.
Same for Table B2
*AC: We changed salinity back to 2 digits. We removed the "rounded up" to prevent confusion. It only meant that some of the values were rounded to 2 digits.*

L424 The data policy of Ocean Science and of almost all other serious journals states that the data should be available as FAIR data. The policy (https://www.ocean-science.net/policies/data_policy.html) says: "If the data are not publicly accessible at the time of final publication, the data statement should describe where and when they will appear, and provide information on how readers can obtain the data until then." Where and when will the data be published?
*AC: We included the transect data in the Appendix that all used data are now presented in the study.*

Thank you and best regards
Mario